# Can We Statically Locate Knowledge in Large Language Models? Financial Domain and Toxicity Reduction Case Studies

**Jordi Armengol-Estapé**[♣][*] **Lingyu Li**[◇] **Sebastian Gehrmann**[◇] **Achintya Gopal**[◇]
**David Rosenberg**[◇] **Gideon Mann** **Mark Dredze**[◇][♥]
[◇]Bloomberg ♣University of Edinburgh ♥Johns Hopkins University
jordi.armengol.estape@ed.ac.uk

## Abstract

Current large language model (LLM) evaluations rely on benchmarks to assess model capabilities and their encoded knowledge. However, these evaluations cannot reveal where a model encodes its knowledge, and thus little is known about which weights contain specific information. We propose a method to statically (without forward or backward passes) locate topical knowledge in the weight space of an LLM, building on a prior insight that parameters can be decoded into interpretable tokens. If parameters can be mapped into the embedding space, it should be possible to directly search for knowledge via embedding similarity. We study the validity of this assumption across several LLMs for a variety of concepts in the financial domain and a toxicity detection setup. Our analysis yields an improved understanding of the promises and limitations of static knowledge location in real-world scenarios.

## 1 Introduction

The impressive text generation abilities of large language models (LLMs) arise from complex interactions among billions of parameters. These parameters encode a vast range of knowledge, allowing models to answer closed-book fact-based questions across dozens of domains. LLM evaluations focus on model abilities: what can the model do and what does it know? However, evaluations based on model outputs cannot answer *where* this knowledge is stored in the network.

If a model correctly answers the question "What city in the United States had the first subway?", why does it matter what parameters store the answer "Boston"? First, we may want to know in what domains is the model capable by simply "looking" at the knowledge-storing parameters, which can provide insights about the model's inner working. Second, we may want to edit or remove a

model's knowledge or behavior, e.g., outdated information, offensive terminology, or stereotypes. Removing this information may be more effective than fine-tuning the model not to express it. Third, in the search for better model architectures, we may want to enhance a model's knowledge storage ability. All of these goals and more require knowing where information is stored inside a model.

Previous research on locating knowledge in language models is divided into dynamic and static analyses. Dynamic analyses focus on examining how model activations and outputs change with different inputs to identify where knowledge is stored and how model capabilities function (Vig et al., 2020; Olsson et al., 2022). Instead, we are focused on investigating the possibilities of *static* knowledge location, that is, without any forward or backward passes. While static knowledge location is challenging due to having access to strictly less information than dynamic methods, its potential simplicity makes it interesting for practical reasons and scientific curiosity.

One static approach to examine the weights in a Transformer network (Vaswani et al., 2017) is to project the model parameters into the word embedding or vocabulary space for interpretation (Elhage et al., 2021; Geva et al., 2022b). Dar et al. (2022) developed the idea that all Transformer parameters, including all Multi-Layer Perceptron (MLPs) and attention layers, can be interpreted by projecting them into the vocabulary space. They also use the embedding space interpretation to align parameters across models based on their vector similarity. The advantage of this approach is that it operates directly on the model parameters without requiring specific inputs or a forward pass.

Rather than projecting all parameters into the vocabulary space, or aligning them based on their vector similarity, we posit that if the embedding space interpretation of the parameters holds, then we should be able to directly locate specific knowl-

---

*Work done during an internship at Bloomberg.

edge given a query in the embedding space. We propose a straightforward method based on embedding similarity that identifies what knowledge is contained within a model and where that knowledge is located without any forward pass. We take two real-world case studies, information extraction tasks focusing on the financial domain and a toxicity reduction setup, and run our experiments at scale (up to 176B parameters). We study the relevance of the parameters identified by our method to the target knowledge, and investigate how specific these locations are by measuring the downstream performance on seemingly unrelated tasks when ablating the found parameters. Finally, we utilize the method to gain insights into how internal model representations vary across layers and how distributed they are, and to have a better understanding of the possibilities and limits of static knowledge location.

## 2 Statically locating parameters

Our goal is to statically identify where knowledge is stored within the parameters of a large language model. We assume a running example of knowing the names of CEOs for companies. This information is part of the financial domain, and is a specific type of information (CEO relation) that applies to many different companies. What parameters in the model store the identities of these CEOs? Popular approaches to locating this information use a forward pass through the model with different inputs to measure how the outputs or activations of the model vary, e.g. by inspecting attention weights (Vashishth et al., 2019; Clark et al., 2019) or gradients (Dai et al., 2022). However, input-based approaches require forward and/or backward passes, which are computationally expensive and may not generalize beyond the tested inputs. Instead, we are interested in locating parameters *statically*, that is, without input, forward or backward passes by building on recent methods of *static* interpretation of Transformers (Geva et al., 2022c; Dar et al., 2022).

These methods can directly interpret model parameters in the embedding space. We represent the model parameters in embedding space, formulate a task-relevant query in the same embedding space, and use it to search over the parameters. For example, to locate parameters containing CEO identities, we take the query "CEO" and search for relevant parameters in the embedding space.

We describe our method in several steps: how the model parameters are interpreted into the embedding space, how a query is represented in this same space, and efficient search of the parameter space.

### 2.1 Interpreting parameters in embedding space

We begin by interpreting the model parameters in the semantic embedding space. Transformers consist of two blocks: a multi-layer perceptron and a self-attention module, which are applied consecutively with residual connections. Elhage et al. (2021) note that this sum of the output of all the previous layers and the original embedding could be understood as a shared communication channel among layers, referred to as the *residual stream*. This property can be exploited to project intermediate outputs into the vocabulary space (nostalgebraist, 2020; Din et al., 2023). In other words, *activations* across the model seem to be in the same embedding space as the input embeddings, $E$, which in most implementations share weights with the language modeling head ($E^T$).

However, we need the *parameters*, and not only the activations, to be in a shared embedding space. Building upon Geva et al. (2022c, 2020) and Elhage et al. (2021), Dar et al. (2022) proposed extending the residual stream view by projecting Transformer *parameters* into the original vocabulary space. We propose using this same insight to represent the Transformer's parameters in the same embedding space as text queries, allowing us to directly (semantically) compare a query with model parameters.[1] In summary, we will identify which model parameters contain names of CEOs by finding those that are most semantically similar to "CEO".

We describe the procedure for both types of model parameters: MLPs and attention.

#### 2.1.1 MLPs

Prior work has shown that MLP blocks function as key-value memories, allowing the Transformer to store knowledge (Geva et al., 2020). The first layer of the MLP block is parameterized (omitting biases) by the weight matrix $W_{in} \in \mathbb{R}^{D' \times D}$, the "keys" of the "memory", where $D$ is the embedding dimension and $D'$ is the hidden dimension of the MLPs. Similarly, the second layer of

---

[1]A similar idea was proposed by Dar et al. (2022) to align parameters across models through vector similarity.

the MLP block is parameterized by a weight matrix $W_{out} \in \mathbb{R}^{D \times D'}$, the "values". In Geva et al. (2022b), the embedding space interpretation of those weights is that each $W_{out}$ column can be seen as an embedding vector in the same space as tokens. Geva et al. (2022c) extended this view to the parameters in the first layer; each row in $W_{in}$ can be seen as an embedding vector in the same space as the tokens. We refer to these parameters as MLP-K (key) and MLP-V (value).

### 2.1.2 Attention

Attention blocks are associated with contextual processing rather than knowledge storage, though previous work has been able to statically interpret these parameters (Dar et al., 2022; Millidge and Black, 2022). Attention blocks are parameterized (omitting biases) by 3 kinds of parameters for each attention head $i \in \{1, 2, ...N\}$: $W_Q^i$, the queries' projection; $W_K$, the keys' projection; and $W_V$, the values' projection. Additionally, there is a shared attention output projection, $W_O$, which can also be split as separate $W_O^i$ for each head. Dar et al. (2022) propose the *subhead view*, which forms embedding space interpretations for the individual units in $W_Q$, $W_K$, $W_V$, and $W_O$ analogously to those we saw for MLPs.

Dar et al. (2022) made assumptions to theoretically justify this embedding interpretation. For instance, they omit biases and layer normalization, and approximate the inverse of $E$, needed for extending the embedding space interpretation to the first layer of MLPs and attention subheads, with $E^T$. We keep these choices since they empirically work well in Dar et al. (2022). In our work, we don't need to explicitly use the inverse as we do not project the parameters to the vocabulary space.

### 2.2 Queries

We now have an interpretation of the different transformer parameters in the embedding space $E$. We cast a given text query $e$ in the same space to compute semantic similarity. We assume that our query effectively represents a knowledge type, e.g., the token "CEO" represents the CEO relation, and that this token is either directly present in $E$, or by pooling multiple tokens present in $E$. Given this embedding query $e \in \mathbb{R}^D$, we retrieve the $k$ Nearest Neighbors (k-NN) over all layers for a specific parameter type (layers in MLP or attention blocks) by returning the projected parameter indices that

maximize the cosine similarity for the query embedding. For example, for the first layer in MLPs:

$$Q(e) = \texttt{topk}\left\{s(\mathbf{e}, \mathbf{p}) \mid \mathbf{p} \in \bigcup_{i=1}^{L} \text{rows}(W_{in}^i)\right\},$$

where $s$ is the cosine similarity and $L$ is the number of layers. We posit that the most similar parameters $\mathbf{p}$ contain knowledge relevant to $\mathbf{e}$.

### 2.3 Implementation

We deal with massive models with tens or hundreds of billions of parameters. Efficiently searching through this space for the parameters most similar to a query is not an easy task. Large models can be loaded into memory efficiently using model parallelism (Shoeybi et al., 2019; Rasley et al., 2020), and we need to rearrange the Transformer parameters to match the embedding interpretation and allow for efficient search. As we saw earlier, the embedding space interpretation for the first layer of MLPs is that each row in $W_{in}$ can be seen as an embedding vector in the same space as tokens. Thus, the shape of the weights already matches its embedding interpretation. However, $W_{out} \in \mathbb{R}^{D \times D'}$ requires transposition to correspond to the embedding shape. We refer by *unit* to the individual weight vectors interpreted in the shape required by the embedding space interpretation, be it a row or a column of the row of the matrix weights depending on the embedding space interpretation of each parameter kind. Similarly, the attention weights require rearrangement corresponding to the subhead view. We refer to the appendix for additional implementation details.

## 3 Experiments

We evaluate the effectiveness of our static search method in identifying parameters that contain knowledge related to the given query. Unlike typical LLM benchmarks, we do not know the "right answer" nor can we evaluate our search in terms of accuracy. Therefore, we develop probes and metrics to measure the extent of the relevant knowledge contained in the identified parameters. We focus on domain-specific knowledge by searching for several types of financial information across several models.

We measure model performance at knowledge tasks under ablation experiments, where we zero

out parameters identified by our method as containing relevant knowledge. Specifically, we eliminate the top-K parameter units identified as being the most similar to the given query, e.g., "CEO" for CEO knowledge questions. We increase the number of ablated units ($k$ corresponds to 0.1%, 0.2%, etc. of the all units) to measure the performance degradation on the knowledge task. We establish baseline performance by comparing to random ablations (see Appendix D.)

We expect that ablating parameters will hurt a model; we seek to show that ablating specific parameters removes specific knowledge. Therefore, we also report performance on control tasks unrelated to the query for which the ablations should have little effect. For example, we query the model about national capitals, unrelated to our financial queries. Additionally, we select this task since it relies on general domain knowledge, for which even small LLMs are likely to do well. We explore the robustness of our results to this choice of control task in Appendix E.

### 3.1 Setup

We consider several different model sizes and families: GPT2-medium (355M) (Radford et al., 2019), OPT-{1.3B, 6.7B, 66B} (Zhang et al., 2022), GPT-NeoX (20B) (Black et al., 2022), and Bloom 176B (BigScience). We also consider two instruction-tuned models based on OPT-1.3B: OPT-IML-1.3B and OPT-IML-MAX-1.3B (Iyer et al., 2023).

We consider several tasks that rely on different types of financial information:

- CEOs: A dataset of 500 CEO-company pairs (S&P 500).The model is asked for the name of the CEO of a given company in a zero-shot setting.

- Tickers: Same as CEOs but asking for stock tickers for a company.

- Ticker extraction (NER-ED): The ticker extraction task in Wu et al. (2023), in which models must extract the tickers of the named entities (companies) appearing in the text.

- Authors: A dataset of best-selling fiction-author pairs.[2] The model is asked to reply with the author name of a given work.

---

[2]https://w.wiki/7Lhr

- Directors: A dataset of Academy Awards movie-director pairs.[3] The model is asked to reply with the director of a given movie.

- Arithmetic: `add_sub_multiple` subset in the test split of the Deepmind Mathematics dataset (Saxton, 2019).

## 4 Results

We summarize our main findings for select models, and include additional results for all tasks and models in the Appendix. In preliminary experiments, we found that knowledge localization was more accurate for MLPs; we thus focused on MLP Ks and Vs for most experiments.

### 4.1 CEO Task

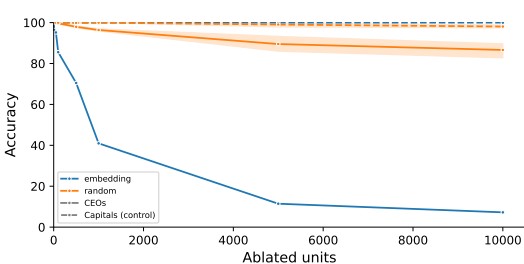

Figure 1: OPT-6.7B accuracy wrt. MLP-Ks ablations, on the CEOs task. Blue lines represent the accuracies with increasing ablated units using the embedding weight localization; orange lines represent those with random ablation. Solid lines correspond to the target task accuracies, while dashed lines correspond to the control task. We can see that the solid blue line decreases faster than both the lines corresponding to the control task and the random ones.

Figure 1 shows the accuracy of the CEOs task as we ablate an increasing number of units (MLP Ks) for OPT-6.7B. The accuracy drops sharply when the closest units in embedding space to "CEO" are removed (solid blue line), while the control task (dashed blue line, Capitals task) remains largely unaffected. Orange lines show accuracy when an equal number of randomly selected units are ablated, which has a significantly smaller effect on the performance. This shows that our method can statically identify parameters responsible for storing relevant knowledge with a certain degree of specificity.

Figure 2 shows the corresponding layer-wise distribution of the ablated units (close to the embedding of "CEO" in embedding space), with darker

---

[3]https://w.wiki/7Li5

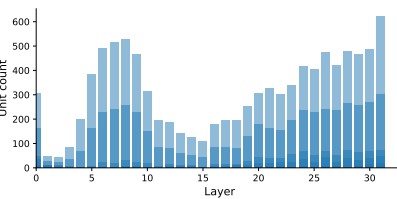

Figure 2: OPT-6.7B layer-wise distributions of ablated MLP-Ks, on the CEOs task. Darker bars correspond to smaller $k$s when locating the top-K closest units. We can see that while there is a certain bias towards the last layer, the overall distribution is far from exhibiting a trivial pattern in which only the units close to the last layer are feasible to locate, with an interesting peak in the first third of the model.

blue indicating fewer units selected. While we observe a strong effect in the first and last layers, the overall pattern is far from trivially selecting the units of the layers close to the embedding layers. A peak between layers 5 and 10 hints at some early processing of the concept of "CEO".

We find similar trends for the CEO task across all models[4]. Table 1 reports the differences between target and control accuracies. We confirm the difference between random and control ablations and the targeted ablation to be positive in all cases. Table 2 shows a compact version of the accuracy plots and histogram we saw before, this time for a selection of the models (OPT-{6.7B, 66B}, GPT-Neox-20B, Bloom-176B) on the CEOs task. For all models, the accuracy on the CEOs task drops sharply, while the accuracy on the control task remains less affected.

## 4.2 Influence of model size and families

**How does model size influence knowledge localization?** Figure 3 shows accuracy in target (solid lines) and control (dashed lines) tasks as more units close in embedding space are removed. We first consider the CEO task with MLP-K ablations. Across all model scales (from 355M to 176B) and families (GPT2, OPT, GPT-Neox, Bloom), there is an early, sharp drop in the target task performance. In contrast, the control task remains largely unaffected until significantly more weights are ablated. For the CEO task, at the extremes, Bloom-176B seems most impacted by ablation (i.e., the largest gap between dashed and solid lines), while GPT2-

---

[4]For comparing the performance of different ablated models, we use the percentage of ablated units over the total model units rather than unit counts since the total model units vary by different model size.

355M is the model least affected by the targeted ablation. Other than this observation, we see no correlation between model size and knowledge localization effectiveness; the technique works similarly across model families and sizes.

**How do other model features affect knowledge localization?** Figure 3 also shows instruction-tuned variants (OPT-IML). Interestingly, their accuracy curves have very similar shapes, and for the target task, the less fine-tuned the model, the lower the accuracy seems to drop, while a seemingly reverse pattern seems to hold for the control task. In Table 1, we observe that in the case of targeted and random ablation on the target task (B-A), the instruction-tuned versions of OPT-1.3B present very similar numbers across ablation levels. In short, instruction-tuned models from the same base model show very similar behaviors when ablated compared to the original base model. Instruction tuning does not change where knowledge is stored or our ability to locate it. We further observe that the control task for GPT-Neox is disproportionately affected. This could be explained by the effect of post-Kaplan (Kaplan et al., 2020) scaling laws used in its pre-training compute. Longer pre-training may lead to less sparse representations, a hypothesis we seek to explore in future work.

## 4.3 Generalization to other tasks

Figure 3 presents results for other tasks: Tickers and Directors. The significant delineation between target and control task shown in Figure 3 for CEOs and Directors tasks suggest that the effectiveness of localization depends on the specificity of the query token. However, the more challenging task of Ticker Extraction also led to promising results, especially with OPT-66B MLP-V (last row in Table 3). We refer to Appendix C for interesting results for the Arithmetic task, despite ablating the seemingly innocuous token "0". Knowledge localization worked for both OPT-66B and Bloom-176B, but less well for GPT-Neox, on which the tickers extraction task didn't show promise either. Overall, GPT-Neox layerwise distributions hint at a more significant signal in the first (MLP-Ks) and last (MLP-V) layers than in the rest of the models.

**What parameters store knowledge?** Examining which parameters were selected confirms that we did not trivially select units from the first and last layer, where the representations are closer to the embedding layer. It also provides insights into

| Model | Target Task | | Control Task | | Accuracy Differences | | |
|---|---|---|---|---|---|---|---|
| | Targeted Ablation (A) | Random Ablation (B) | Targeted Ablation (C) | Random Ablation (D) | (B) - (A) | (C) - (A) | (D) - (A) |
| GPT2-355M | 76.67 | 97.69±3.17 | 94.04 | 98.49±1.82 | 21.02±3.17 | 17.37 | 21.82±1.82 |
| OPT-1.3B | 78.42 | 97.37±1.74 | 99.85 | 99.24±0.40 | 18.96±1.74 | 21.43 | 20.82±0.40 |
| OPT-IML-1.3B | 79.23 | 97.51±2.55 | 98.41 | 99.34±0.30 | 18.28±2.55 | 19.18 | 20.11±0.30 |
| OPT-IML-Max-1.3B | 77.79 | 96.62±1.08 | 98.48 | 99.15±0.53 | 18.83±1.08 | 20.69 | 21.36±0.53 |
| OPT-6.7B | 69.05 | 97.88±0.67 | 100.00 | 99.90±0.23 | 28.83±0.67 | 30.95 | 30.85±0.23 |
| GPT-Neox-20B | 35.10 | 85.96 | 89.67 | 97.18 | 50.86 | 54.57 | 62.08 |
| OPT-66B | 75.20 | 99.21 | 81.82 | 81.82 | 24.01 | 6.62 | 6.62 |
| Bloom-176B | 46.17 | 92.31 | 100.00 | 100.00 | 46.14 | 53.83 | 53.83 |

Accuracy given 0.1% Ablation

| Model | Target Task | | Control Task | | Accuracy Differences | | |
|---|---|---|---|---|---|---|---|
| | Targeted Ablation (A) | Random Ablation (B) | Targeted Ablation (C) | Random Ablation (D) | (B) - (A) | (C) - (A) | (D) - (A) |
| GPT2-355M | 41.38 | 83.12±4.43 | 88.43 | 92.97±2.36 | 41.74±4.43 | 47.05 | 51.59±2.36 |
| OPT-1.3B | 18.50 | 86.80±2.51 | 98.12 | 97.37±0.99 | 68.30±2.51 | 79.62 | 78.87±0.99 |
| OPT-IML-1.3B | 15.16 | 84.22±3.94 | 97.15 | 96.02±1.14 | 69.06±3.94 | 81.99 | 80.86±1.14 |
| OPT-IML-Max-1.3B | 12.01 | 85.60±5.51 | 96.03 | 97.10±0.94 | 73.59±5.51 | 84.02 | 85.10±0.94 |
| OPT-6.7B | 7.23 | 86.63±4.48 | 100.00 | 98.09±0.97 | 79.40±4.48 | 92.77 | 90.86±0.97 |
| GPT-Neox-20B | 13.60 | 75.88 | 70.78 | 96.01 | 62.28 | 57.18 | 82.41 |
| OPT-66B | 23.72 | 90.45 | 81.82 | 81.82 | 66.73 | 58.10 | 58.10 |
| Bloom-176B | 15.00 | 87.31 | 99.23 | 99.78 | 72.31 | 84.23 | 84.78 |

Accuracy given 2.0% Ablation

Table 1: Results on the CEOs task with MLP-Ks. We show accuracies at different ablation levels (0.1% ablated units, 0.5% ablated units, etc). For each ablation level, for each model, we report 4 accuracies: A) the targeted ablation results (i.e., ablating units close to "CEO") on the target task (CEOs task), B) the random ablation results on the target task, C) the targeted ablation results on the control task (Capitals task), and D) the random ablation results on the control task. We also report the differences between these accuracies. We expect B-A and C-A to be positive, which means that the random ablation has less effect on the performance than the targeted ablation on the target task, and that the control task performance is less affected than the target task, respectively.

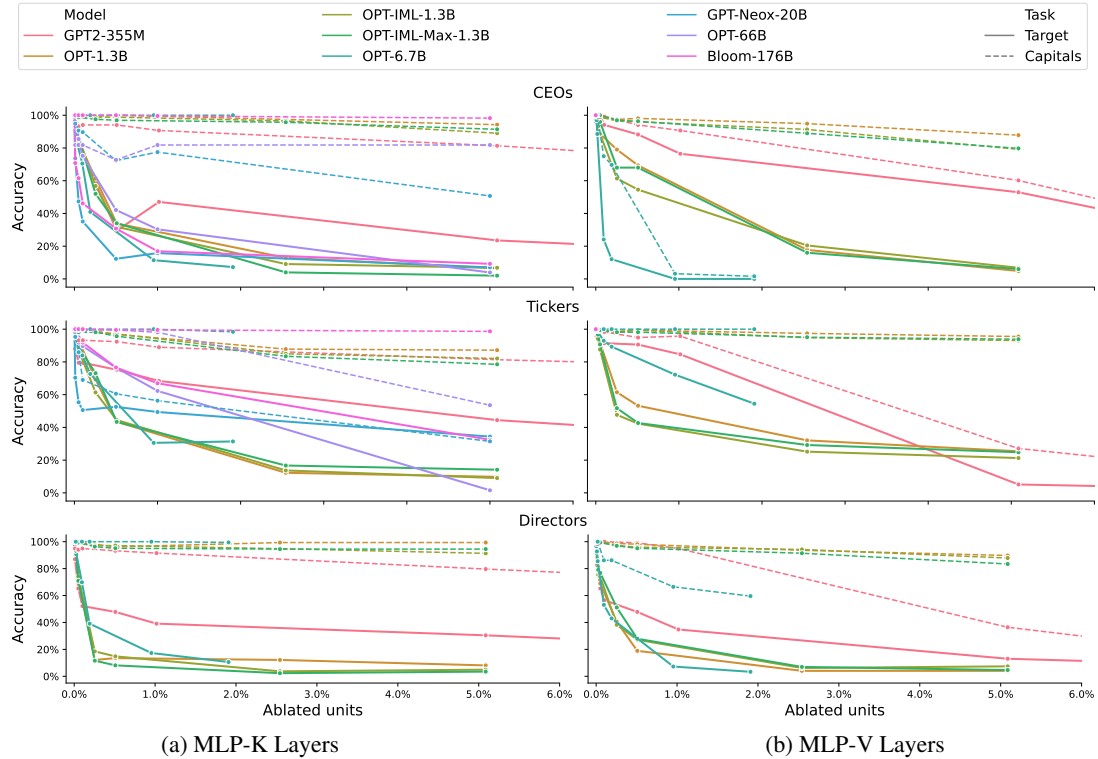

Figure 3: Accuracy for the CEOs, Tickers, and Directors tasks when ablating using the embedding location method, for both the target tasks (CEOs, Tickers, and Directors) and the control task (Capitals). We can see how the target (solid) lines are generally below the control (dashed) lines, as expected, across model scales and architectures, especially for the CEOs and Directors tasks.

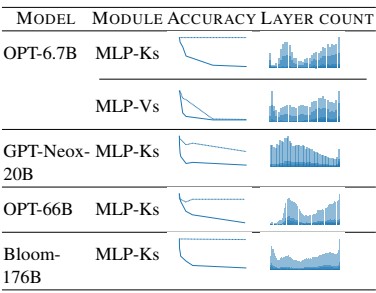

| MODEL | MODULE | ACCURACY | LAYER COUNT |
|---|---|---|---|
| OPT-6.7B | MLP-Ks | | |
| | MLP-Vs | | |
| GPT-Neox-20B | MLP-Ks | | |
| OPT-66B | MLP-Ks | | |
| Bloom-176B | MLP-Ks | | |

Table 2: Results on the CEOs task, including accuracies for the target and control task when using the weight location method, and the layer-wise unit distribution.

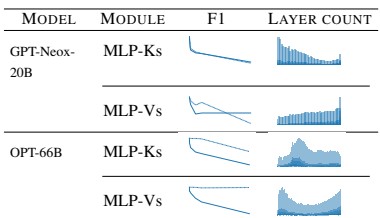

| MODEL | MODULE | F1 | LAYER COUNT |
|---|---|---|---|
| GPT-Neox-20B | MLP-Ks | | |
| | MLP-Vs | | |
| OPT-66B | MLP-Ks | | |
| | MLP-Vs | | |

Table 3: Results on the Ticker Extraction task.

how the model stores and processes knowledge. Tables 2, 3, and 5 also present the histograms of the layer unit counts. The overall trend, similar to the findings in Vig et al. (2020) and nostalgebraist (2020), is for the unit density in either the first or last layers. More interestingly, some models show distinctive fingerprint-like patterns. For example, all OPT results with MLP-K ablation have a peak around the first third together with another peak at the last layer, especially in the larger variants. OPT results with MLP-Vs ablations follow a U-shape distribution. In other model families, GPT-Neox units concentrate on the first layers with MLP-K ablations and the last layers with MLP-V ablations.

**How well can knowledge be localized in different parameter types?** As mentioned at the beginning of section 4, we found that knowledge localization was more accurate for MLPs. We still include Attention results in the Appendix, as shown in Tables 6. We successfully select Attention weights (as in, the target task performance drops faster than control) in about half the cases, but less reliably than in the case of MLPs. This is consistent with the fact that a) we evaluated knowledge-intensive tasks, and b) prior work (Geva et al., 2020, 2022b) suggests that MLPs are more involved in this kind of task than attention modules.

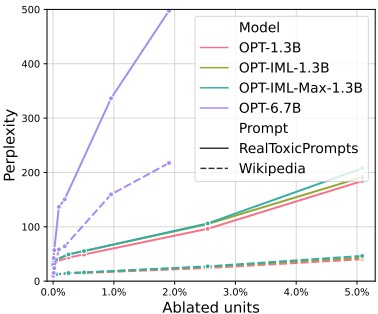

Figure 4: Perplexity check on MLP-V units ablation.

# 5 Toxicity Reduction

Our evaluation so far has been on model knowledge: can we identify where information is stored? We now turn to model behavior: can we identify what model parameters are responsible for toxic language generation?

LLMs can generate toxic text that contains offensive language and biased beliefs and stereotypes (Gehman et al., 2020). Several strategies exist to mitigate these generations during inference (Dathathri et al., 2020), remove behavior during fine-tuning (Liu et al., 2021), filter pretraining data to remove biases (Zhang et al., 2022), and neuron-level interventions that edit the model parameters directly (Geva et al., 2022c; Li et al., 2023).

We adopt a strategy similar to the neuron-intervention methods and use our technique to identify and ablate model parameters associated with toxic generation. We apply our method to OPT 1.3b, OPT 6.7b, OPT IML 1.3b, and OPT IML MAX 1.3b and measure the reduction in toxicity. Unlike prior methods that locate toxic units by projecting them into the vocabulary space (Geva et al., 2022c) or by learning ablation masks from fine-tuning models on the toxic dataset (Li et al., 2023), we hypothesize that toxicity can be removed by ablating parameters found using our static knowledge localization method. We form a query to represent a toxic concept by averaging the embeddings of 24 toxic tokens. We locate the K nearest units to this concept embedding and ablate them as we did in our above evaluations. We measure the proportion of toxic outputs generated from each ablated model (following Hanu and Unitary team, 2020), as well as the perplexity of 2,000 prompts sampled from RealToxicityPrompts (Gehman et al., 2020) and Wikipedia (Foundation). Additional details on the methodology and evaluation are in Appendix F.

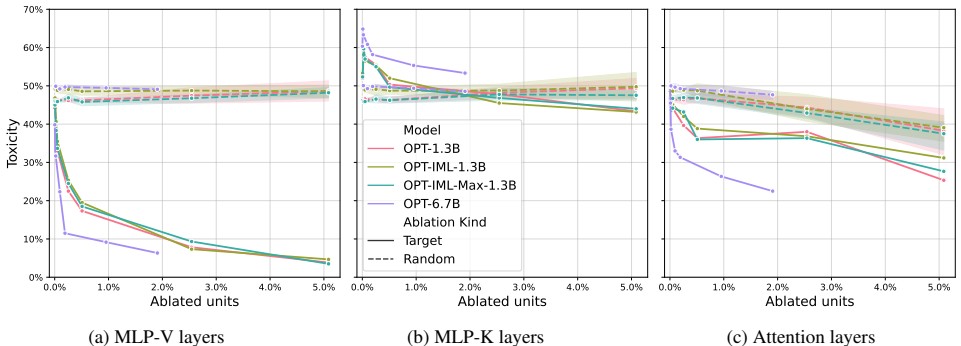

(a) MLP-V layers      (b) MLP-K layers      (c) Attention layers

Figure 5: Results of toxicity unit ablation studies where each subplot shows the effectiveness of the ablation method given three different types of model layer. Each subplot compares the unit ablation method (KNN) against a random ablation baseline across four different models. The ablated units in the x-axis represent the percentage of ablated units over the total model units. The toxicity in the y-axis represent the percentage of the toxic model output given 600 toxic prompts.

Figure 5 shows that the ablation of MLP-V units reduces the toxicity of the models, with a drop of more than 35% when only 1% of the units are ablated. Ablating the same number of random units leads to unchanged toxicity. Curiously, ablating MLP-K units has no significant effect on toxicity, and ablating Attention units reduces the toxicity of the models, but not as effectively as MLP-V's.

**Ablating MLP-V units increases the toxic language modeling perplexity with minor impairment of the generic one.** Figure 4 shows that, when ablating ~1% of MLP-V units on models with size 1.3B, the perplexity of model generations on toxic prompts increases from 31.81 to 49.03 (+17.22), while the perplexity of model generation on non-toxic text (Wikipedia) increases from 11.19 to 15.07 (+3.88). The MLP-V units' ablation impedes the ability of the model to model toxic language while having a smaller effect on the overall language modeling performance. However, for OPT 6.7b, the degradation in perplexity for Wikipedia is significant, increasing from 8.62 to 64.12 (+55.5), implying that the scale of the model is inversely correlated with the sparsity of units that encode broad concepts related to language modeling, or that this methodology of defining a toxicity embedding does not scale to larger models.

**Ablated toxicity MLP-V units are distributed around the early layers.** Table 4 shows the MLP-V layer distribution of the ablated units. Surprisingly, all models demonstrate that toxicity related units tend to be concentrated in the early layers, which is rarely seen in other ablation tasks. The early layers are though to be associated with shal-

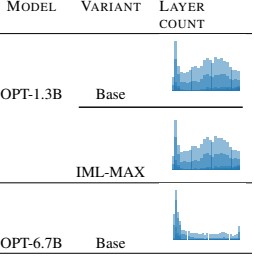

| MODEL | VARIANT | LAYER COUNT |
|---|---|---|
| OPT-1.3B | Base | |
| | IML-MAX | |
| OPT-6.7B | Base | |

Table 4: MLP-V's Layer Count on Toxicity

low patterns (Geva et al., 2020).

## 6 Related work

Previous work on locating knowledge in neural networks falls into two broad categories: dynamic and static analyses. Dynamic analyses are concerned with model activations. Given multiple inputs, these methods look at how activations change, and thus deduce where knowledge is stored or how model capabilities function. To this end, Vig et al. (2020) propose using causal mediation analysis to investigate Transformer behaviors and apply it to gender bias. In the follow-up work, Finlayson et al. (2021) shows that the same technique can be used to locate syntactic phenomena such as subject-verb agreement. Similarly, Meng et al. (2023) and Meng et al. (2022) propose using causal interventions to locate and edit factual knowledge.

Static analyses focus on model weights directly. Our work builds on a line of research that projects model parameters into an interpretable space (Geva et al., 2020, 2022b; Elhage et al., 2021; Dar et al., 2022). Geva et al. (2022a) investigated keyword search over Transformer parameters, although their work is limited to the second layer of MLPs, and

search over the tokens projected from the parameters, rather than directly on the parameters themselves.

## 7  Conclusions

We demonstrated that by casting the parameters of an LLM into embedding space and directly performing embedding similarity search with respect to a query, we can localize stored knowledge without a forward pass. We have studied the performance after ablating the selected parameters and the layer-wise distribution of these parameters in two real-world settings on a diverse range of models to gain insights on the promises and limits of static knowledge location.

## Limitations

In this work, we have not included evaluations for more recent models such as Llama (Touvron et al., 2023). We base our work on Dar et al. (2022), which does not support SwiGLU (Shazeer, 2020) out of the box (due to the added parameters).

Additionally, the evaluations are limited to domain-specific tasks, and control and target tasks are not necessarily equally easy to ablate. We explore the robustness of our results to this choice of control task in the Appendix E.

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

# A  Background

In this section, we review the fundamental ideas in Vaswani et al. (2017) and Dar et al. (2022) required to better understand our proposed solution from a technical standpoint.

## A.1  Transformer

Transformers (Vaswani et al., 2017) are mainly characterized by the following hyperparameters[5]:

- $D$, embedding/hidden size

- $|V|$, vocabulary size

- $L$ layers

- $N$ attention heads

- $D^n = \frac{D}{N}$ head dimension

- $D^l$ = hidden dimension of MLPs (typically $4D$)

**Embedding and unembedding**  As a first step in the Transformer, each input token from the discrete sequence is embedded into a real-valued vector through the embedding matrix $E \in R^{D \times |V|}$. Each token $t$ is embedded as follows:

$$x = \texttt{embed}(t) = E[:, t_{id}]$$

with $x \in \mathbb{R}^D$ and where $t_{id}$ is the token id (index) corresponding to the token $t$. This maps tokens to the *embedding space* of vectors in $\mathbb{R}^D$. Some models also apply layer normalization after the embedding layer, but we omit it here.

Similarly, an *unembed* (or language modeling head) layer is used as the last step, to go back to the *vocabulary space* of logits in $\mathbb{R}^{|V|}$. Typically, it's also a linear transformation with a weight matrix $U \in \mathbb{R}^{|V| \times D}$:

$$\texttt{unembed}(x) = Ux$$

Again, some models apply layer normalization in unembed but we omit it here. Since $U$ has the transposed dimensions of E, many implementations tie embedding and unembedding layers to the same layers (e.g., GPT-2).[6]  In this work, we assume that the unembed layer is linear and tied to the embedding layer, which is a realistic assumption for most models, with $U = E^T$

---

[5]While we focus on decoder-only language models, our method is not restricted to this kind of Transformers.

[6]Other works such as Devlin et al. (2018) opt for a non-linear unembedding layer.

**Transformer layer**  After the embedding layer,[7] a decoder-only model is composed of N identical layers. Let $X \in \mathbb{R}^{T \times D}$ be a sequence of embedded tokens, with $T$ being the sequence length, and $x$ be an individual embedded token (real-valued vector with dimension $D$) in the sequence. Each layer has the following structure:

$$\texttt{Layer}(X) = \texttt{MLP}\{\texttt{LN}[\texttt{MHA}(\texttt{LN}(X)) + X]\} + X$$

where MLP stands for Multi-Layer Perceptron (MLP) and MHA stands for Multi-Head Attention. The exact order of application of LN (Layer Norm) varies across implementations.

**Multi-Head Attention**  Transformers mix information from the token embeddings in a given sequence with pairwise dot-product multi-head attention. We will first see how each attention head is (independently) defined for each attention head, and later we will see how the head outputs are aggregated. The first step is projecting the token representations in the input sequence $X$ into Queries (Q), Keys (K), and Values (V) as follows:

$$Q = XW_Q + b_q; K = XW_K + b_k; V = XW_V + b_v$$

$$Head(X) = \texttt{softmax}\left(\frac{QK^T}{\sqrt{D^k}} \odot M\right)V$$

where $b_{\{q,k,v\}}$ are the bias terms and $\odot M$ is the element-wise multiplication by the masking matrix, defined as:

$$M = \left[m_{i,j}\right]_{i,j} \in \mathbb{R}^{T \times T}$$

$$m_{i,j} = \mathbb{1}(i \leq j) - \infty \cdot \mathbb{1}(i > j) \quad \forall i, j \in [T]$$

with $\mathbb{1}(a)$ being an indicator function returning 1 if the predicate $a$ is true and 0 otherwise. This element-wise multiplication by $M$ has the effect of creating a causal, triangular attention mask that prevents leaking future token information.

In decoder models, Q, K, and V come all from the input sequence, attending to itself. We omit unmasked self-attention or cross-attention due to our focus on decoder-only models.

The outputs of each head $Head_n(X) \forall n \in [N]$ are then concatenated and projected back to the embedding dimension:

$$[\texttt{Head}_n(X) \forall n \in [N]] W_o + b_o$$

---

[7]Some models also have a positional embedding in this step, which we omit here.

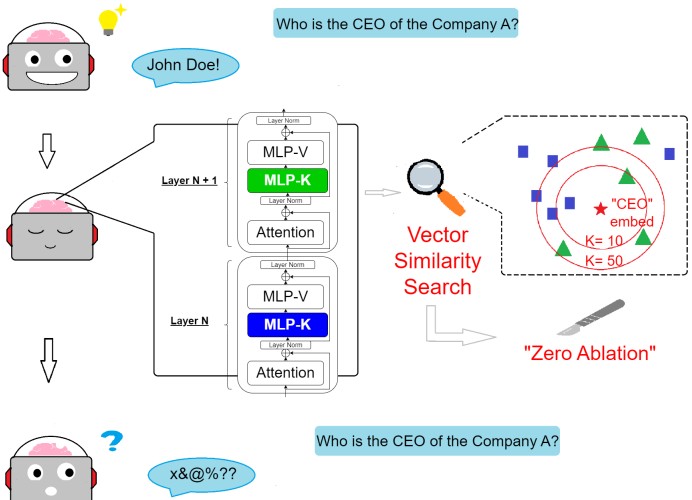

Figure 6: Overview of our approach: Interpreting Transformer weights in the embedding space as in Dar et al. (2022) allows us to perform k-NN embedding similarity search over the parameters given an embedding corresponding to a task-relevant token of the model's embedding matrix. Then, we can intervene by ablating the retrieved units.

**MLPs** MLPs are independently applied element-wise to each $x \in X$:

$$\texttt{MLP}(x) = \texttt{act}(XW_{in} + b_{in})W_{out} + b_{out}$$

where $act$ is the activation function (typically GELU (Hendrycks and Gimpel, 2016)) and $b_{\{in,out\}}$ are the bias terms. The first layer parameters ($W_{in}$) can be referred to as keys and the second layer ones ($W_{out}$), as values, following the findings in Geva et al. (2020) that Transformer MLPs act as Key-Value memories, but they are not to be confused with the keys and values of the attention block.

**Layernorm** Layernorm (LN) is a normalization function that is applied to hidden states of the Transformer before/after (depending on the implementation) attention and MLP blocks:

$$LN(x) = \frac{x - \mu(x)}{\sqrt{\sigma^2(x) + \epsilon}} \odot \gamma + \beta$$

where $\gamma$ and $\beta \in \mathbb{R}^D$ are learnable parameters, $\epsilon$ is a constant value added to the denominator for numerical stability. and $\mu$ and $\sigma$ are computed independently for each token in the sequence $x \in \mathbb{R}^D$:

$$\mu(x) = \frac{1}{D} \sum_i x_i \in \mathbb{R}$$

$$\sigma^2(x) = \frac{1}{D} \sum_i (x_i - \mu(x))^2 \in \mathbb{R}$$

Note that each individual LN (the different ones in each layer and across different layers) has an independent parameterization of $\gamma$ and $\beta$.

**Residual stream** Both MLP and attention blocks are applied as residual connections, meaning that the output of these blocks is summed to the previous hidden states. Elhage et al. (2021) noted that this sum of the output of all the previous layers and the original embedding could be understood as a shared communication channel among layers. Taking this view to the extreme, hidden states should be able to be projected into the original vocabulary space.

We omit dropout (Srivastava et al., 2014) and positional encodings since their presence and specifics vary across implementations.

### A.2 Embedding space interpretation of parameters

There is a vast literature on Transformers' interpretability from different lenses. In this work, we are interested in the view that Transformer parameters can be analyzed in the embedding space (Dar et al., 2022). Geva et al. (2022c) showed that the values (second layer in MLPs) of Transformers can be interpreted in the embedding space. Elhage et al. (2021) showed that attention parameters could be interpreted in embedding space in small models. Dar et al. (2022) generalized those previous findings to all Transformer parameters (both attention and MLPs). In the remainder of this section, we summarize the parameter projections proposed in

Geva et al. (2022c) and Dar et al. (2022).

A matrix $M \in \mathbb{R}^{m \times D}$ can be projected into the vocabulary space by matrix multiplication with the embedding matrix $E \in \mathbb{R}^{D \times |V|}$, yielding $M' \in \mathbb{R}^{m \times |V|}$. Each of the rows in $M'$ represents the affinity between the embedding vector and each vocabulary item, and the `argmax` would yield the most probable token. Following the residual stream view, nostalgebraist (2020) apply these vocabulary projections to the Transformer hidden states to observe how the model progressively builds up its final token predictions. Geva et al. (2022c) and Dar et al. (2022) go further and claim that these projections to the vocabulary layer can be applied directly to the MLP and attention *parameters* (rather than activations) yielding arguably interpretable neurons.

More specifically, the matrix $M$ can correspond to a weight matrix. For interpreting a row vector of $M$, $v$, Geva et al. (2022c) (corresponding to the weights of an individual unit) follow two steps. The first one is the projection to project $v$ into the vocabulary space, $vE$. The second step is to take the `top-k argmax` $vE$, and these `top-k` tokens would correspond to the tokens the most related to the unit parameterized by $v$. Geva et al. (2022c) posits that this interpretation is sound since the most activated vector coordinates contribute the most when added to the residual stream.

Geva et al. (2022c) can only apply this method to the *values* of MLPs (weights of the second layer), because these are the ones directly being added to the residual stream. Dar et al. (2022) posit that inner products and matrix multiplications in a Transformer can be interpreted in the embedding space if we assume a right inverse of $E$, $E'$, such that we can approximately reconstruct the original matrix.

### A.2.1 MLPs

MLPs blocks have been shown to work as key-value memories where the Transformer stores knowledge (Geva et al., 2020), so we expect to be able to locate knowledge in their parameters. The first layer of the MLP block is parameterized[8] by the weight matrix $W_{in} \in \mathbb{R}^{D' \times D}$, the "keys" of the "memory". Similarly, the second layer of the MLP block is parameterized by a weight matrix $W_{out} \in \mathbb{R}^{D \times D'}$, the "values". In Geva et al. (2022b), the embedding space interpretation of those weights is that each $W_{out}$ column can be

---
[8]Omitting biases.

seen as an embedding vector in the same space as tokens.

Geva et al. (2022c) extended this view to the keys (first layer of MLPs) as follows. According to the view of Transformer MLPs as Key-Value memories (Geva et al., 2020), with $x$ being the hidden state input to the MLP (the "queries" to the memory):

$$x W_{in}^T = x E E' W_{in}^T = x E E' W_{in}^T = x E (W_{in} E'^T)^T$$

Assuming the residual stream interpretation, according to which $xE$ should be interpretable in the vocabulary pace, then $W_{in} E'^T$ should also be interpretable in the vocabulary space since they directly interact through an inner product in the MLPs' "memory". Thus, each row in $W_{in}$ can be seen as an embedding vector in the same space as the tokens. Finally, note that $W_{in} E'^T$ can be approximated as $W_{in} E$ s

### A.2.2 Attention

Omitting biases, attention blocks are parameterized by 3 kinds of parameters for each head $i \in \{1, 2, ... N\}$: $W_Q^i$, the queries' projection; $W_K$, the keys' projection; and $W_V$, the values' projection. Additionally, there is a shared attention output projection, $W_O$, that can also be split as separate $W_O^i$ for each head (the part of the projection matrix interacting with the corresponding head after the concatenation).

Dar et al. (2022) consider two possibilities for projecting these weight matrices into the vocabulary space, namely, the interaction matrices, and the subheads view.

**Interaction matrices**    Elhage et al. (2021) proposed interpreting attention through the interaction matrices of queries-values, $W_{QK}$, and values-output projection $W_{VO}$. From the dot-product attention formula, it's easy to see that If we omit biases and define $Q = X W_Q$ and $K = X W_K$, it's easy to see that $W_Q$ and $W_K$ interact directly and in an input-independent way when computing the dot product:

$$Q K^T = X W_Q (X W_K)^T = X W_Q W_K^T X$$

Similarly, we can see that $W_V$ and $W_O$ interact directly after the concatenation of the different head outputs. All in all, for each head $i$ we can define:

$$W_{QK}^i = W_Q^i W_K^{i\,T} \in \mathbb{R}^{D \times D}$$

$$W_{\text{VO}}^i = W_V^i W_O^i \in \mathbb{R}^{D \times D}$$

Similarly to what we saw for MLPs' keys, we can now follow how (Dar et al., 2022) interpret these matrices in the embedding space. Like MLPs' values, the output of the attention block is directly added to the residual stream and, thus, we expect it to be meaningfully projected into the embedding space. With a reasoning analogous to what we saw in the case of an MLP, making use of $E^I$ and interpreting inner products in the embedding space, Dar et al. (2022) showed that the other parameter kinds ($W_Q$, $W_K$, and $W_V$) can also be approximately projected into the vocabulary space in a meaningful way.

**Subhead view** Dar et al. (2022) propose an alternative view to the attention interaction matrices that has the advantage of being able to project individual units. Using the identity $AB = \sum_{j=1}^{b} A_{:,j} B_{j,:}$:

$$W_{\text{VO}}^i = \sum_{j=1}^{\frac{D}{N}} W_V^{i,j} W_O^{i,j}, \quad W_{\text{QK}}^i = \sum_{j=1}^{\frac{D}{N}} W_Q^{i,j} W_K^{i,j\ \text{T}}$$

This allows for the definition of *subheads*. Subheads are the vector columns of $W_Q^i, W_K^i, W_V^i$, that is, $W_Q^{i,j}, W_K^{i,j}, W_V^{i,j} \in \mathbb{R}^{D \times 1}$, respectively. They can be approximately projected to the vocabulary space by multiplication by $E$. Additionally, the row vectors $W_O^{i,j} \in \mathbb{R}^{1 \times D}$ of f $W_O^i$ are also subheads, and they can be directly projected to the vocabulary space by multiplication by $E$ without any approximation.

### A.3 Other parameters

Layer-norm is ignored in this approach, following Elhage et al. (2021)'s observation that it can be ignored because normalization changes only magnitudes and not the direction of the update. Biases and the effects of positional encoding are also omitted in this approach for the sake of simplicity.

Finally, we note that Dar et al. (2022) propose using $E^T$ as an approximation to the right inverse $E^I$ due to a) being a good enough approximation, and b) yielding more interpretable results. However, in our case, since we never need to project parameters to the vocabulary space, we sidestep the need for directly using this inverse approximation.

## B Additional technical details

### B.1 Implementation details

Storing rearranged weights with a list of tensor views (one list for each parameter kind, each element in the list being a tensor view corresponding to the weight matrix of a given layer and parameter kind), rather than creating a new tensor with all the parameters, allows to store the reshaped weights' data as a reference to the original one. This has the benefits of a) decreasing memory overhead, b) keeping the original device sharding in case of LLMs, and c) being able to directly modify the model weights by modifying the rearranged ones.

Each model architecture requires its own weight loader, since weight storage varies across implementations (e.g., the GPT-2 family implements the MLP as a 1-D convolution layer, meaning that the weights are transposed; in some implementations, the attention keys, queries, and values projections are stored as a single linear layer). Finally, we note that optimizations typically employed in k-NN settings would be directly applicable here.

### B.2 Experimental settings

In all cases, we study parameter kinds separately: $W_{in}$, $W_{out}$, and attention (for attention, we separately select the top K units for each among the 4 parameter - kinds, queries, keys, values and output projection - and ablate all of them at once). We use the simplest ablation method by setting the corresponding weights to zero to validate the hypothesis that the selected weights are related to the erased concept in the most extreme case. Zeroing out the weights has an indirect effect on the general layer statistics, which might explain the drop in performance of the control task after a significant amount of (presumably unrelated) weights have been ablated.

In all cases, we use a fixed set of numbers for setting K when conducting an experiment on a given model, i.e. 10, 50, 100, 500, 1k, 5k, 10k. However, the total number of units that a model contains varies from one model to another. For easing the comparison of different models in plots, we transform the K values to the proportions over the total number of units that a given model has. For reporting the ablated models' accuracy in tables, we apply interpolations on each model's accuracy given their transformed proportions of ablated units to achieve the same set of proportions of ablated units, i.e. we select points 0.1%, 0.5%, 1.0%, 2.0%.

# C  Arithmetic task

Table 5 summarized the results on the Arithmetic task when ablating using "0" as query.

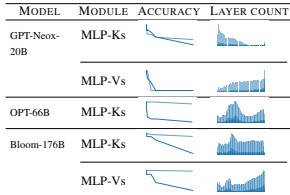

| MODEL | MODULE | ACCURACY | LAYER COUNT |
|---|---|---|---|
| GPT-Neox-20B | MLP-Ks | | |
| | MLP-Vs | | |
| OPT-66B | MLP-Ks | | |
| Bloom-176B | MLP-Ks | | |
| | MLP-Vs | | |

Table 5: Results on the Arithmetic task.

## D Random ablation

In all cases, we conduct the random ablations as a comparison to the targeted ablation. We use a random number generator to pick the K units randomly and ablate them. For the small models with < 10B parameters, we use 5 different random seeds to run random ablations for 5 times and use their mean accuracy and standard deviation for the plots and tables. For the big models with > 10B parameters, we only use one seed to run random ablation for the plots and tables.

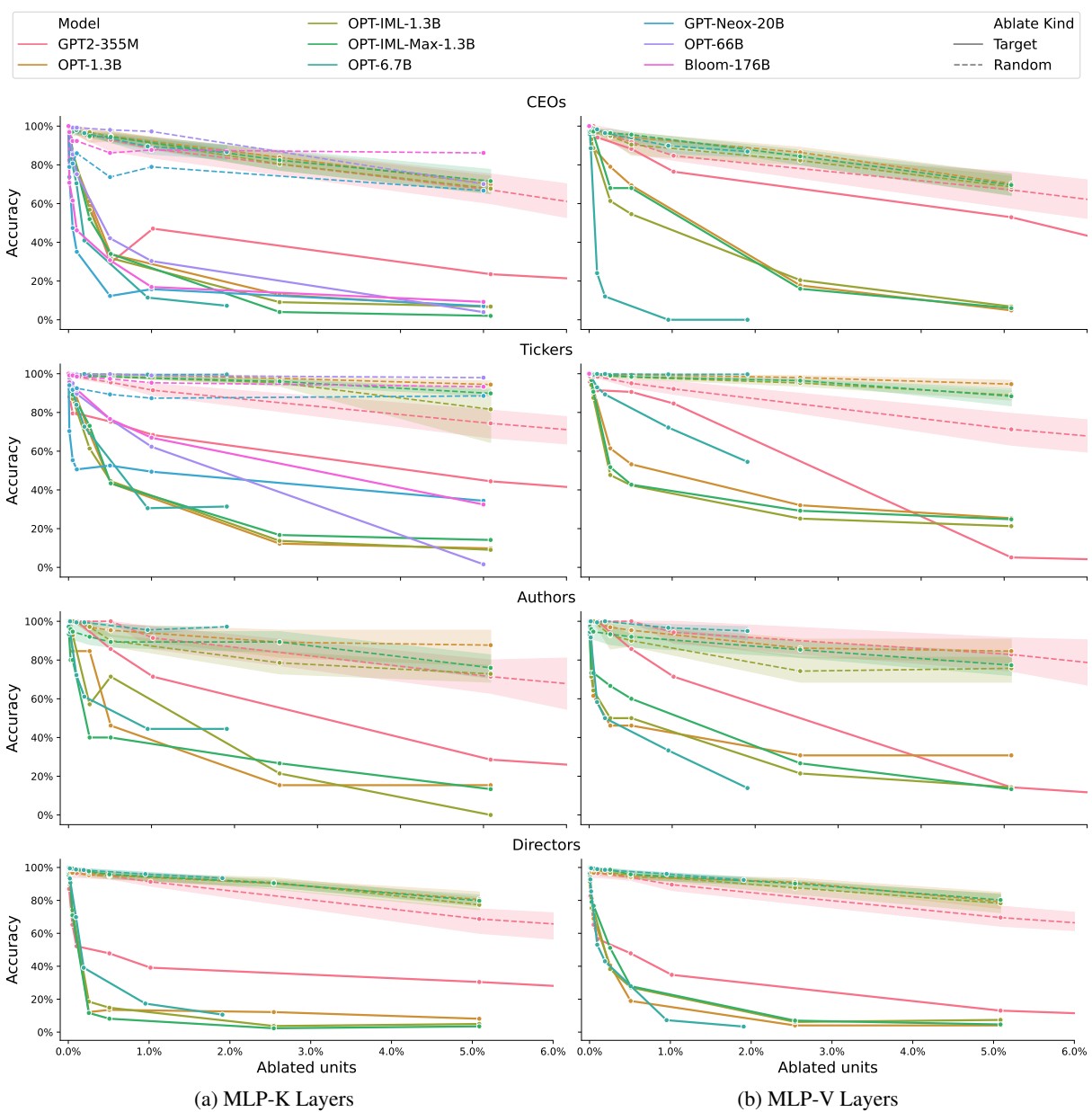

Figure 7: Accuracy on all the tasks against random ablation

## E    Control Group

### E.1    "Capitals" task vs Control group

In the main sections, we use the "Capitals" task as a control task to evaluate how specific our method can ablate the target task's knowledge embedding, that is, reducing the target task accuracy while not reducing the control task's. In this section, we use more control tasks to investigate the robustness of our results when only the "Capitals" control task is used. For any given target task, we compare the accuracy drop of the "Capitals" task as opposed to a set of control tasks(i.e., the control group). For example, in Figure 8, we compare the results on the "CEOs" target task where we use the "Capitals" as our control task (Figure 8a) against where we use the control group including the "Capitals", "Tickers", "Directors" and "Authors" tasks (Figure 8b). We plot the control group using its members' mean accuracy and the standard deviation. We repeat this analysis for the "Tickers", "Directors", and "Authors" target tasks.

As shown in Figure 8, 9 10 and 11, we find: On the "CEOs" target task, the control group mean accuracy drops about 30% - 40% on average when ablating about 2% units on models MLP-K layers, while the "Capitals" control task accuracy drops about 0% - 10%. This implies that our ablation method affects control tasks differently. However, at 2% units ablation on the MLP-K layers, the control group's largest accuracy drop (about 50%) is less than the target task's accuracy drop (about 80% - 90% for the most models, 60% for GPT2-335M), which shows our ablation method is still effective on locating and ablating the knowledge parameters. While figures of the "Tickers", "Directors" or "Authors" target task report different numbers for the above comparison, the above implication still applies to these target tasks.

### E.2    Control group analysis

To further understand why our ablation method affects the control group's tasks differently, we compare the control tasks in the control group given a target task's ablation on the MLP-K and MLP-V layers. Similar to the Appendix E.1, we iterate this analysis over all four target tasks. As shown in Figure 12, we find: On the "CEOs" target task at 2.0% units ablation on MLP-K layers, the least accuracy drop (about 10%) is from the "Capitals" control task while the largest accuracy drop (about 50%) is from the "Tickers" control task. On the "Tickers"

target task at 2.0% units ablation on MLP-K layers, the least accuracy drop (about 15%) is still from the "Capitals" control task while the largest accuracy drop (about 70%) is from the "CEOs" control task. This implies that the "CEOs" and "Tickers" tasks are more correlated than the rest of the tasks while the "Capitals" task is more independent than other tasks. We observe the similar behavior between the "Authors" and "Directors" tasks when used as the target tasks.

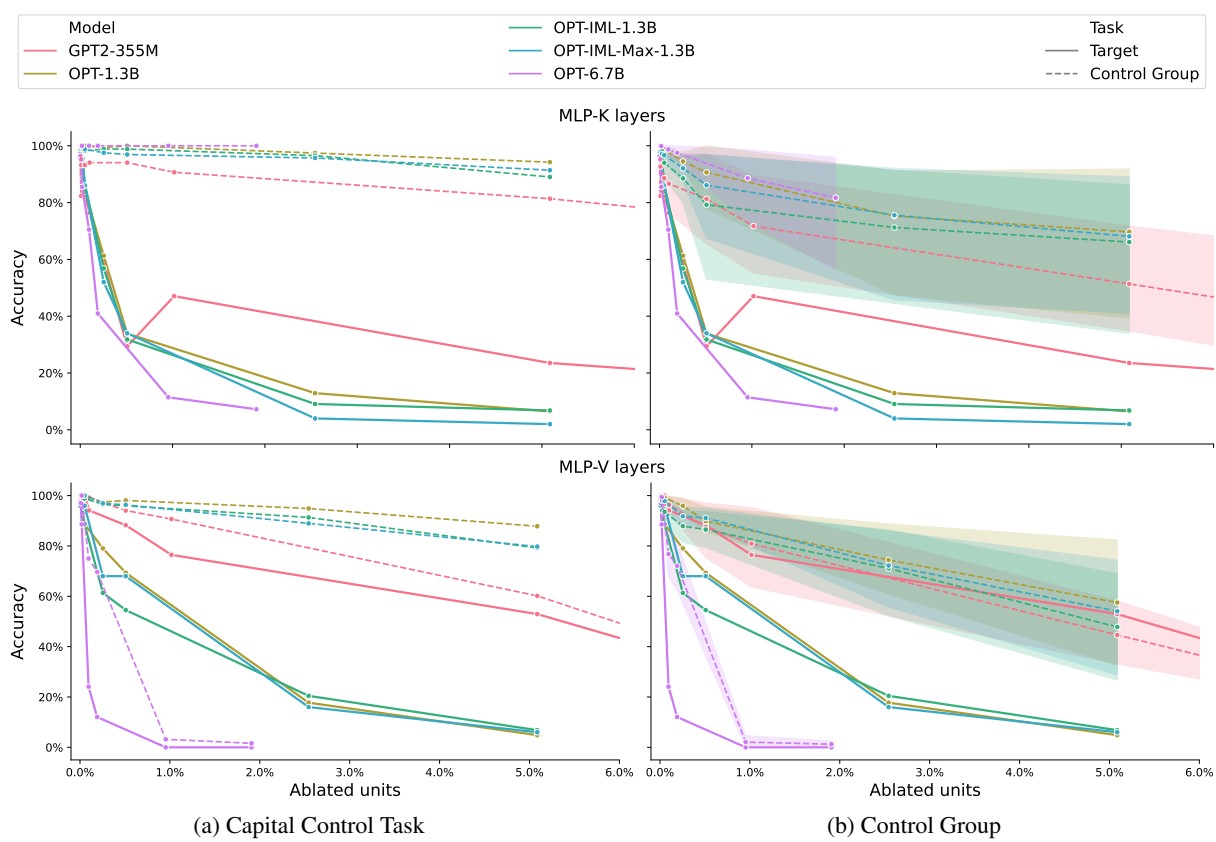

(a) Capital Control Task

(b) Control Group

Figure 8: Models' accuracies on the CEOs task. Control Group Tasks include: Capital, Ticker, Director and Author

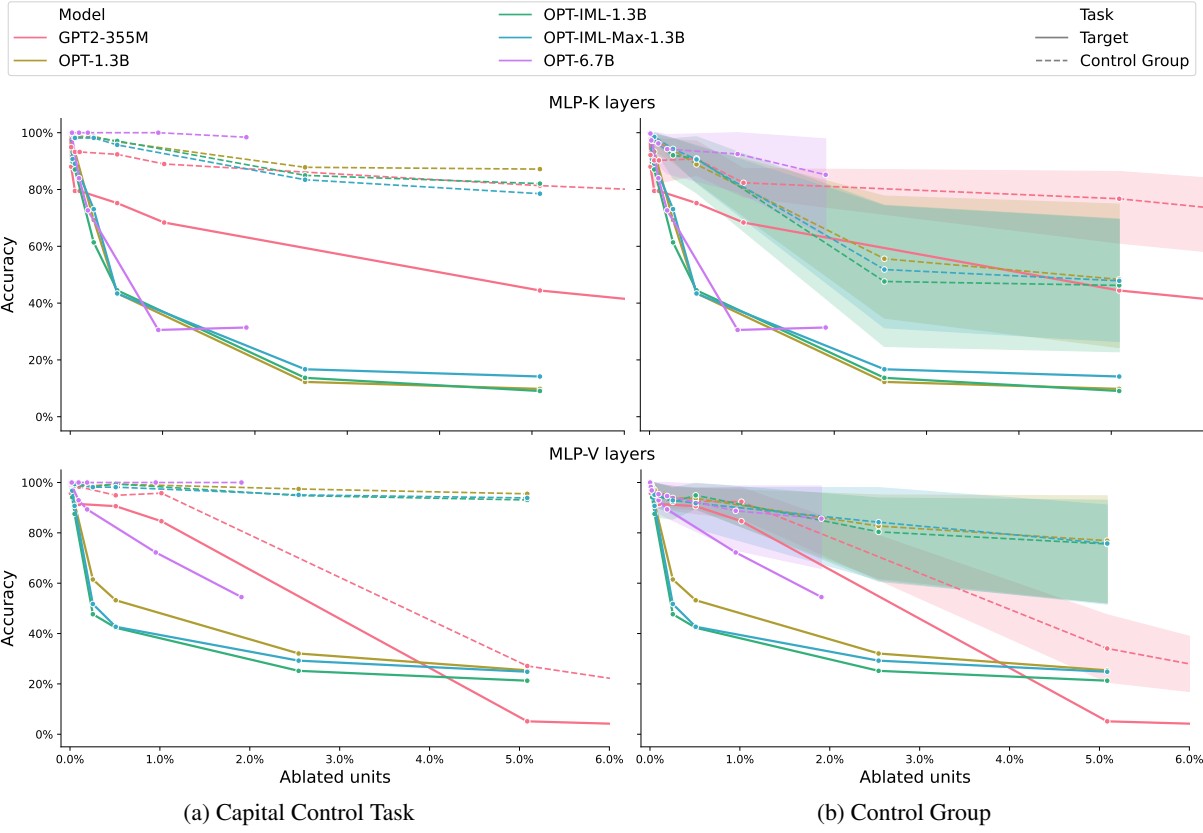

(a) Capital Control Task

(b) Control Group

Figure 9: Models' accuracies on the Tickers task. Control Group Tasks include: Capital, CEO, Director and Author

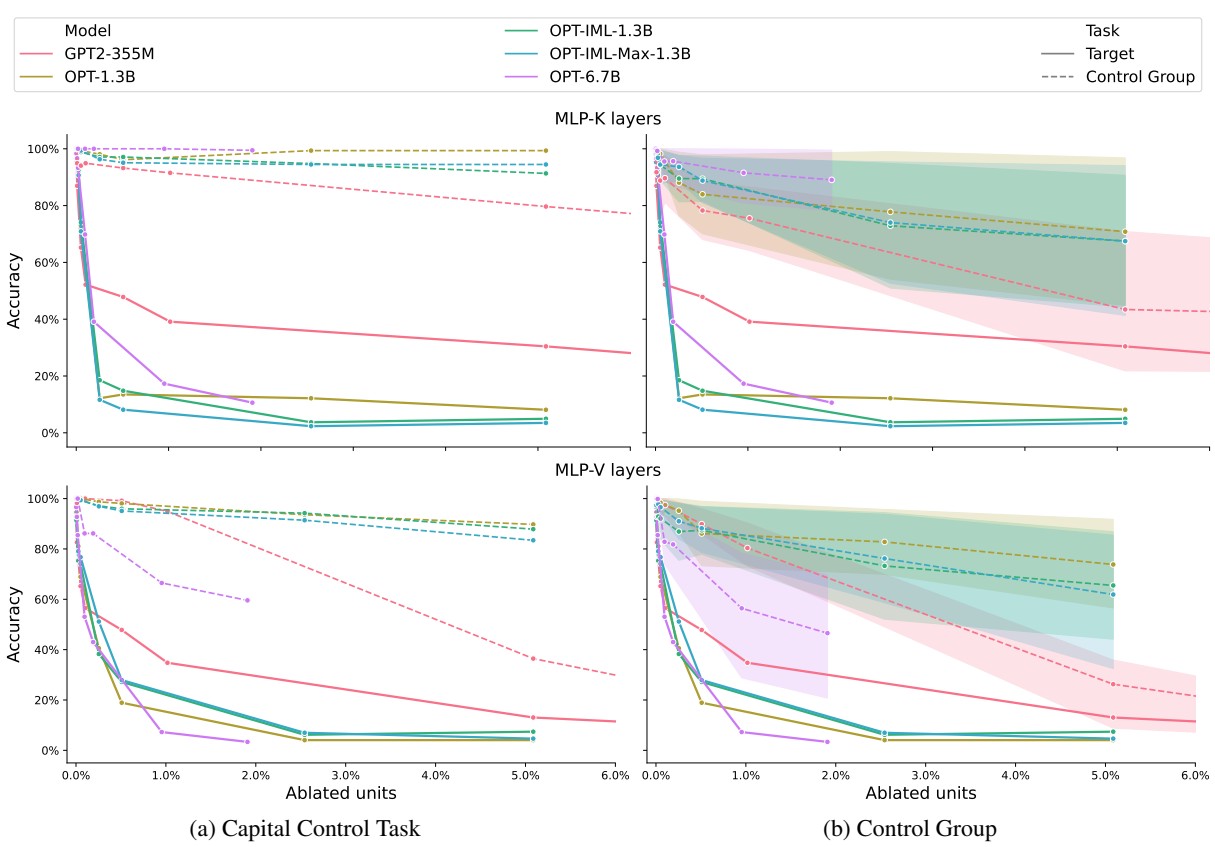

(a) Capital Control Task

(b) Control Group

Figure 10: Models' accuracies on the Directors task. Control Group Tasks include: Capital, CEO, Ticker and Author

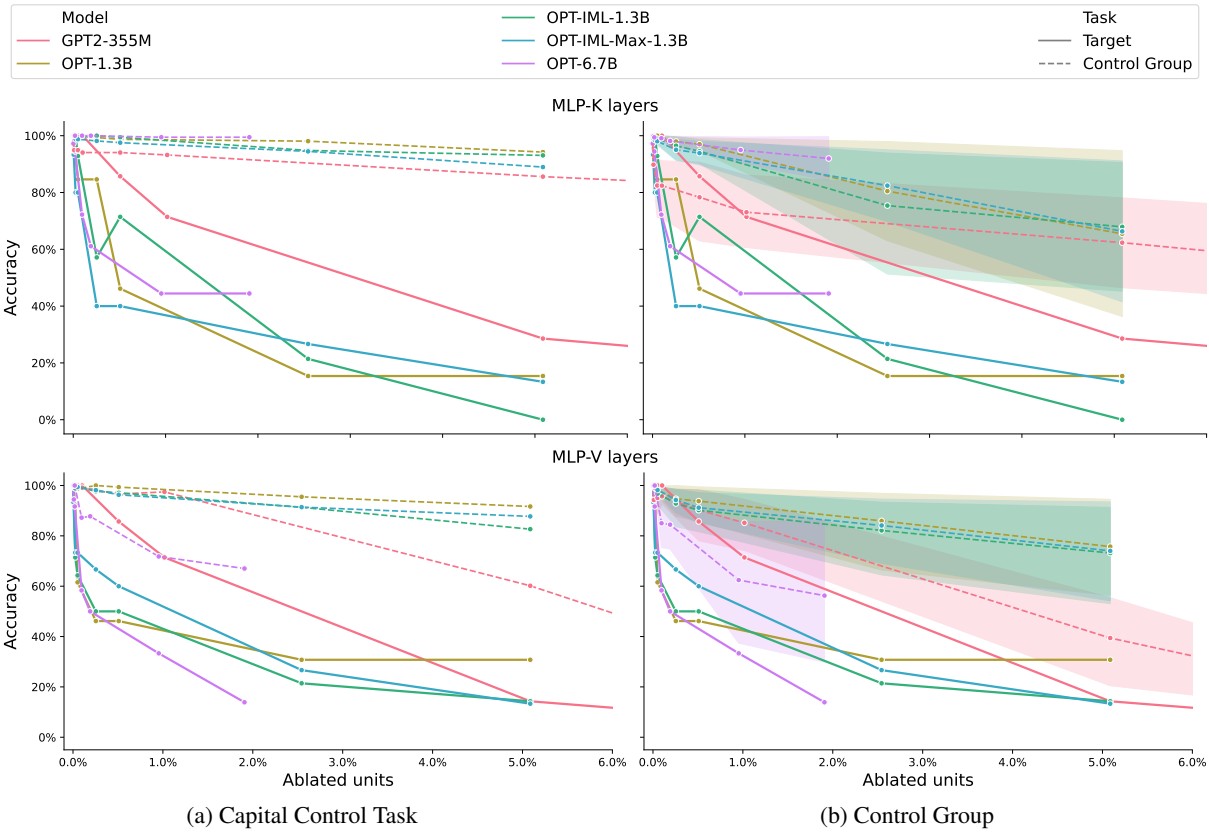

(a) Capital Control Task

(b) Control Group

Figure 11: Models' accuracies on the Authors task. Control Group Tasks include: Capital, CEO, Ticker and Director

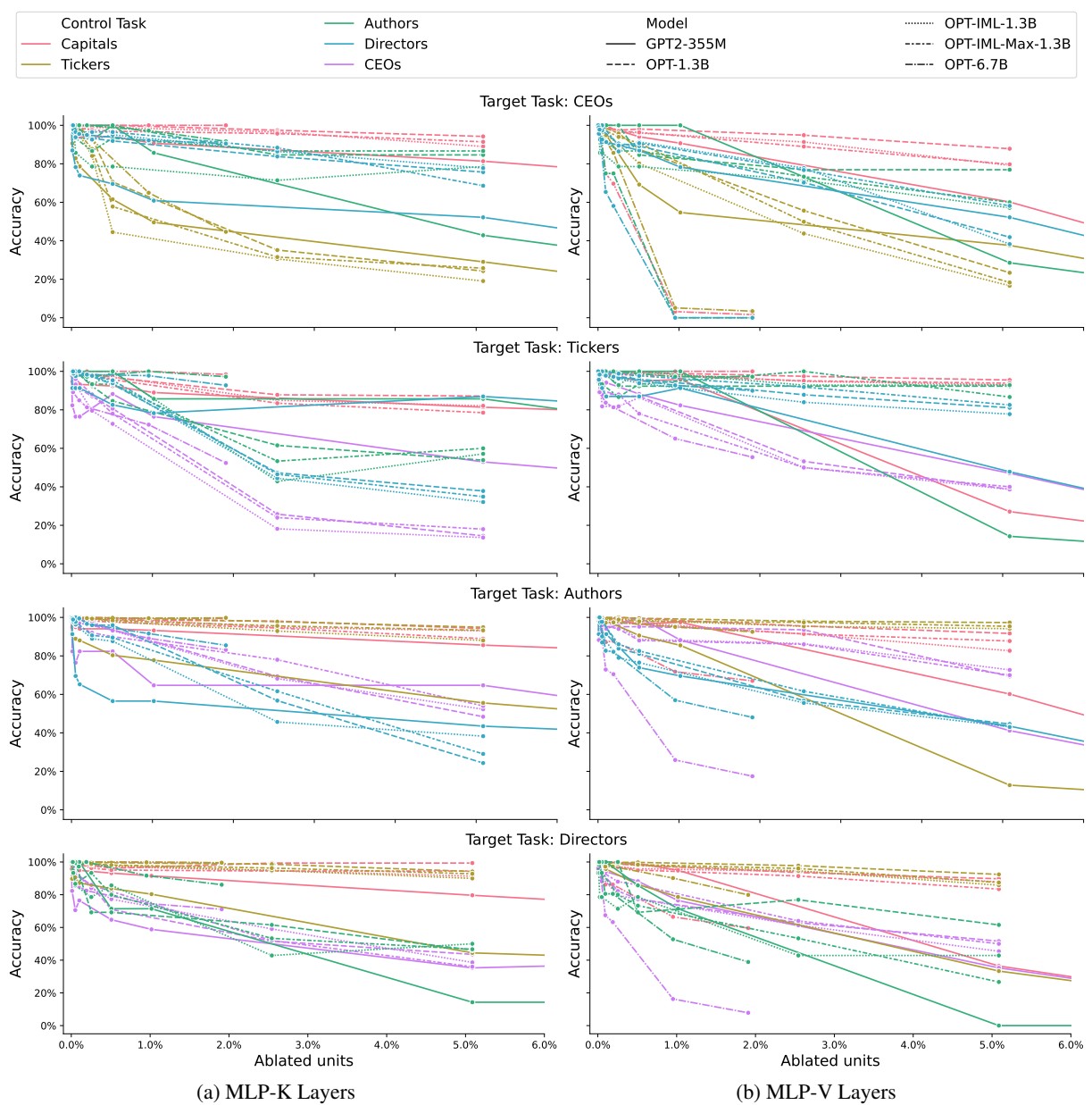

(a) MLP-K Layers

(b) MLP-V Layers

Figure 12: Accuracy on all the tasks in the control group

# F Toxicity reduction task

## F.1 Method.

Inspired by the topic knowledge lookup method (Dar et al., 2022), we demonstrate that toxicity can be represented as a topic embedding, which is utilized to locate knowledge neurons. To obtain this topic embedding of toxicity, we follow these steps:

1. Identify the most common toxic keywords by sampling from a commonly-used list of offensive words.[9] This process yields 24 of the most toxic tokens.

2. Retrieve the corresponding embeddings of these 24 toxic tokens from the model's embedding table.

3. Compute the average of these embeddings to obtain the topic embedding of toxicity.

After getting this topic embedding, we retrieve the K nearest neighbors of projected parameters using cosine similarity and zero out these parameters of selected K knowledge neurons. We perform zero-ablations on various types of layer separately: attention, MLP-K, and MLP-V.

## F.2 Evaluation.

We evaluate our method on a subset of RealToxicityPrompts (Gehman et al., 2020), a collection of 600 prompts designed to elicit toxic responses generated from models. We employ Detoxify (Hanu and Unitary team, 2020), a toxicity classifier, to assess the toxicity of a model's output when presented with a prompt. Detoxify provides 6 metrics, and we classify the model's output as toxic if any of the output scores from these 6 metrics are $> 0.5$. Given the total 600 prompts, we evaluate the toxicity of each ablated model by calculating the proportion of toxic outputs generated from each ablated model. To demonstrate the effectiveness of our method, we conduct random ablation experiments with 5 different seeds as a baseline method to compare with our method.

For an additional check, we measure the perplexity of the ablated models when presented with 2000 prompts sampled from RealToxicityPrompts to assess their language generation performance. We also measure the perplexity of the ablated models when presented with another 2000 prompts from Wikipedia (Foundation) as a control task.

# G Additional results

---

[9]List of Dirty, Naughty, Obscene, and Otherwise Bad Words, downloaded from https://github.com/LDNOOBW/List-of-Dirty-Naughty-Obscene-and-Otherwise-Bad-Words

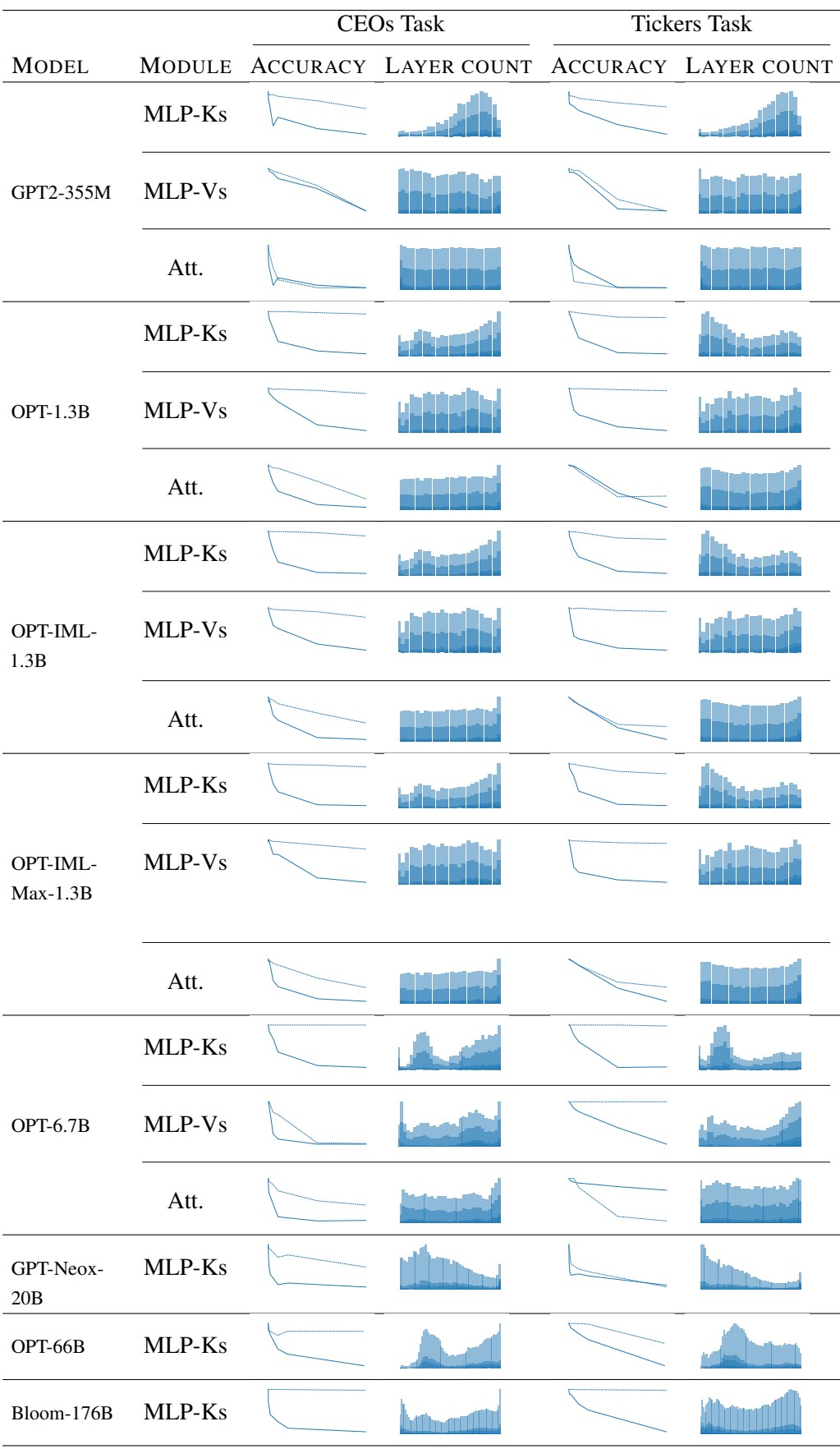

Table 6: Results on the CEOs and Tickers task, including accuracies for the target and control task when using the embedding weight location method and the layer-wise unit distribution.

| Model | Target Task | | Control Task | | Accuracy Differences | | |
|---|---|---|---|---|---|---|---|
| | Targeted Ablation (A) | Random Ablation (B) | Targeted Ablation (C) | Random Ablation (D) | (B) - (A) | (C) - (A) | (D) - (A) |
| GPT2-355M | 76.67 | 97.69±3.17 | 94.04 | 98.49±1.82 | 21.02±3.17 | 17.37 | 21.82±1.82 |
| OPT-1.3B | 78.42 | 97.37±1.74 | 99.85 | 99.24±0.40 | 18.96±1.74 | 21.43 | 20.82±0.40 |
| OPT-IML-1.3B | 79.23 | 97.51±2.55 | 98.41 | 99.34±0.30 | 18.28±2.55 | 19.18 | 20.11±0.30 |
| OPT-IML-Max-1.3B | 77.79 | 96.62±1.08 | 98.48 | 99.15±0.53 | 18.83±1.08 | 20.69 | 21.36±0.53 |
| OPT-6.7B | 69.05 | 97.88±0.67 | 100.00 | 99.90±0.23 | 28.83±0.67 | 30.95 | 30.85±0.23 |
| GPT-Neox-20B | 35.10 | 85.96 | 89.67 | 97.18 | 50.86 | 54.57 | 62.08 |
| OPT-66B | 75.20 | 99.21 | 81.82 | 81.82 | 24.01 | 6.62 | 6.62 |
| Bloom-176B | 46.17 | 92.31 | 100.00 | 100.00 | 46.14 | 53.83 | 53.83 |

Accuracy given 0.1% Ablation

| Model | Target Task | | Control Task | | Accuracy Differences | | |
|---|---|---|---|---|---|---|---|
| | Targeted Ablation (A) | Random Ablation (B) | Targeted Ablation (C) | Random Ablation (D) | (B) - (A) | (C) - (A) | (D) - (A) |
| GPT2-355M | 30.41 | 93.04±6.35 | 94.07 | 96.65±2.20 | 62.63±6.35 | 63.66 | 66.24±2.20 |
| OPT-1.3B | 34.80 | 94.82±3.09 | 99.98 | 98.60±1.11 | 60.02±3.09 | 65.18 | 63.80±1.11 |
| OPT-IML-1.3B | 32.67 | 94.62±3.71 | 98.84 | 98.73±0.93 | 61.96±3.71 | 66.18 | 66.06±0.93 |
| OPT-IML-Max-1.3B | 34.61 | 94.41±2.54 | 96.95 | 97.93±1.02 | 59.80±2.54 | 62.34 | 63.32±1.02 |
| OPT-6.7B | 29.00 | 93.60±1.92 | 100.00 | 99.61±0.51 | 64.60±1.92 | 71.00 | 70.61±0.51 |
| GPT-Neox-20B | 12.29 | 73.69 | 72.31 | 96.24 | 61.40 | 60.02 | 83.96 |
| OPT-66B | 42.13 | 98.03 | 72.73 | 81.82 | 55.90 | 30.60 | 39.69 |
| Bloom-176B | 30.77 | 86.15 | 100.00 | 100.00 | 55.38 | 69.23 | 69.23 |

Accuracy given 0.5% Ablation

| Model | Target Task | | Control Task | | Accuracy Differences | | |
|---|---|---|---|---|---|---|---|
| | Targeted Ablation (A) | Random Ablation (B) | Targeted Ablation (C) | Random Ablation (D) | (B) - (A) | (C) - (A) | (D) - (A) |
| GPT2-355M | 46.46 | 88.39±7.12 | 90.79 | 94.32±2.19 | 41.93±7.12 | 44.33 | 47.86±2.19 |
| OPT-1.3B | 28.81 | 92.19±2.48 | 99.38 | 98.19±1.05 | 63.38±2.48 | 70.57 | 69.38±1.05 |
| OPT-IML-1.3B | 26.33 | 91.14±3.63 | 98.29 | 97.83±0.94 | 64.81±3.63 | 71.96 | 71.51±0.94 |
| OPT-IML-Max-1.3B | 26.75 | 91.50±3.46 | 96.64 | 97.65±0.71 | 64.75±3.46 | 69.88 | 70.89±0.71 |
| OPT-6.7B | 11.24 | 89.38±4.64 | 100.00 | 99.00±1.19 | 78.14±4.64 | 88.76 | 87.76±1.19 |
| GPT-Neox-20B | 15.79 | 78.95 | 77.46 | 96.24 | 63.16 | 61.68 | 80.46 |
| OPT-66B | 30.32 | 97.24 | 81.82 | 81.82 | 66.93 | 51.50 | 51.50 |
| Bloom-176B | 16.92 | 87.69 | 99.56 | 100.00 | 70.77 | 82.64 | 83.08 |

Accuracy given 1.0% Ablation

| Model | Target Task | | Control Task | | Accuracy Differences | | |
|---|---|---|---|---|---|---|---|
| | Targeted Ablation (A) | Random Ablation (B) | Targeted Ablation (C) | Random Ablation (D) | (B) - (A) | (C) - (A) | (D) - (A) |
| GPT2-355M | 41.38 | 83.12±4.43 | 88.43 | 92.97±2.36 | 41.74±4.43 | 47.05 | 51.59±2.36 |
| OPT-1.3B | 18.50 | 86.80±2.51 | 98.12 | 97.37±0.99 | 68.30±2.51 | 79.62 | 78.87±0.99 |
| OPT-IML-1.3B | 15.16 | 84.22±3.94 | 97.15 | 96.02±1.14 | 69.06±3.94 | 81.99 | 80.86±1.14 |
| OPT-IML-Max-1.3B | 12.01 | 85.60±5.51 | 96.03 | 97.10±0.94 | 73.59±5.51 | 84.02 | 85.10±0.94 |
| OPT-6.7B | 7.23 | 86.63±4.48 | 100.00 | 98.09±0.97 | 79.40±4.48 | 92.77 | 90.86±0.97 |
| GPT-Neox-20B | 13.60 | 75.88 | 70.78 | 96.01 | 62.28 | 57.18 | 82.41 |
| OPT-66B | 23.72 | 90.45 | 81.82 | 81.82 | 66.73 | 58.10 | 58.10 |
| Bloom-176B | 15.00 | 87.31 | 99.23 | 99.78 | 72.31 | 84.23 | 84.78 |

Accuracy given 2.0% Ablation

Table 7: Results on the CEOs task with MLP-Ks.

| Model | Target Task | | Control Task | | Accuracy Differences | | |
|---|---|---|---|---|---|---|---|
| | Targeted Ablation (A) | Random Ablation (B) | Targeted Ablation (C) | Random Ablation (D) | (B) - (A) | (C) - (A) | (D) - (A) |
| GPT2-355M | 94.32 | 97.69±3.17 | 99.15 | 98.66±1.52 | 3.37±3.17 | 4.84 | 4.34±1.52 |
| OPT-1.3B | 86.37 | 97.70±1.67 | 99.38 | 99.30±0.42 | 11.32±1.67 | 13.01 | 12.93±0.42 |
| OPT-IML-1.3B | 82.05 | 96.38±2.75 | 98.29 | 99.23±0.43 | 14.33±2.75 | 16.24 | 17.18±0.43 |
| OPT-IML-Max-1.3B | 89.24 | 97.01±1.40 | 99.26 | 99.42±0.33 | 7.77±1.40 | 10.02 | 10.19±0.33 |
| OPT-6.7B | 23.51 | 98.22±0.73 | 74.74 | 99.89±0.22 | 74.71±0.73 | 51.23 | 76.38±0.22 |

Accuracy given 0.1% Ablation

| Model | Target Task | | Control Task | | Accuracy Differences | | |
|---|---|---|---|---|---|---|---|
| | Targeted Ablation (A) | Random Ablation (B) | Targeted Ablation (C) | Random Ablation (D) | (B) - (A) | (C) - (A) | (D) - (A) |
| GPT2-355M | 88.36 | 91.89±6.62 | 94.18 | 96.32±1.62 | 3.53±6.62 | 5.82 | 7.96±1.62 |
| OPT-1.3B | 69.68 | 94.20±2.91 | 98.06 | 98.97±0.56 | 24.52±2.91 | 28.37 | 29.28±0.56 |
| OPT-IML-1.3B | 54.78 | 90.61±6.75 | 95.97 | 98.39±1.35 | 35.83±6.75 | 41.20 | 43.61±1.35 |
| OPT-IML-Max-1.3B | 68.00 | 95.63±1.66 | 96.34 | 98.42±0.54 | 27.63±1.66 | 28.34 | 30.42±0.54 |
| OPT-6.7B | 7.16 | 93.75±1.69 | 42.73 | 99.55±0.51 | 86.58±1.69 | 35.56 | 92.38±0.51 |

Accuracy given 0.5% Ablation

| Model | Target Task | | Control Task | | Accuracy Differences | | |
|---|---|---|---|---|---|---|---|
| | Targeted Ablation (A) | Random Ablation (B) | Targeted Ablation (C) | Random Ablation (D) | (B) - (A) | (C) - (A) | (D) - (A) |
| GPT2-355M | 76.87 | 84.95±3.19 | 90.79 | 93.98±2.60 | 8.08±3.19 | 13.92 | 17.11±2.60 |
| OPT-1.3B | 56.89 | 92.32±2.14 | 97.30 | 98.51±0.34 | 35.43±2.14 | 40.41 | 41.62±0.34 |
| OPT-IML-1.3B | 46.31 | 88.48±5.86 | 94.84 | 97.57±1.26 | 42.17±5.86 | 48.53 | 51.26±1.26 |
| OPT-IML-Max-1.3B | 55.44 | 92.89±2.14 | 94.54 | 97.90±0.51 | 37.45±2.14 | 39.10 | 42.46±0.51 |
| OPT-6.7B | 0.00 | 89.73±4.27 | 3.11 | 99.02±1.21 | 89.73±4.27 | 3.11 | 99.02±1.21 |

Accuracy given 1.0% Ablation

| Model | Target Task | | Control Task | | Accuracy Differences | | |
|---|---|---|---|---|---|---|---|
| | Targeted Ablation (A) | Random Ablation (B) | Targeted Ablation (C) | Random Ablation (D) | (B) - (A) | (C) - (A) | (D) - (A) |
| GPT2-355M | 70.79 | 80.44±3.91 | 83.31 | 92.42±2.67 | 9.66±3.91 | 12.52 | 21.64±2.67 |
| OPT-1.3B | 31.52 | 88.52±2.36 | 95.73 | 97.56±0.44 | 57.00±2.36 | 64.21 | 66.04±0.44 |
| OPT-IML-1.3B | 29.56 | 84.46±4.19 | 92.56 | 95.92±1.12 | 54.90±4.19 | 63.01 | 66.37±1.12 |
| OPT-IML-Max-1.3B | 29.88 | 87.39±3.89 | 90.92 | 96.88±0.75 | 57.51±3.89 | 61.04 | 66.99±0.75 |
| OPT-6.7B | 0.00 | 86.87±3.57 | 1.60 | 98.51±0.69 | 86.87±3.57 | 1.60 | 98.51±0.69 |

Accuracy given 2.0% Ablation

Table 8: Results on the CEOs task with MLP-Vs.

| | Target Task | | Control Task | | Accuracy Differences | | |
|---|---|---|---|---|---|---|---|
| Model | Targeted Ablation (A) | Random Ablation (B) | Targeted Ablation (C) | Random Ablation (D) | (B) - (A) | (C) - (A) | (D) - (A) |
| GPT2-355M | 47.26 | 89.53±8.60 | 83.25 | 96.84±1.23 | 42.27±8.60 | 35.99 | 49.58±1.23 |
| OPT-1.3B | 78.86 | 91.83±3.40 | 96.84 | 98.35±0.72 | 12.97±3.40 | 17.98 | 19.49±0.72 |
| OPT-IML-1.3B | 84.40 | 93.04±4.07 | 95.26 | 97.94±0.62 | 8.64±4.07 | 10.86 | 13.54±0.62 |
| OPT-IML-Max-1.3B | 84.89 | 92.37±2.23 | 94.37 | 98.42±0.18 | 7.48±2.23 | 9.48 | 13.53±0.18 |
| OPT-6.7B | 38.33 | 92.49±1.79 | 84.94 | 99.55±0.42 | 54.16±1.79 | 46.62 | 61.23±0.42 |

Accuracy given 0.1% Ablation

| | Target Task | | Control Task | | Accuracy Differences | | |
|---|---|---|---|---|---|---|---|
| Model | Targeted Ablation (A) | Random Ablation (B) | Targeted Ablation (C) | Random Ablation (D) | (B) - (A) | (C) - (A) | (D) - (A) |
| GPT2-355M | 6.76 | 73.29±11.36 | 47.38 | 93.79±2.41 | 66.54±11.36 | 40.63 | 87.04±2.41 |
| OPT-1.3B | 39.37 | 72.43±14.75 | 92.33 | 96.92±1.13 | 33.06±14.75 | 52.96 | 57.55±1.13 |
| OPT-IML-1.3B | 45.84 | 75.35±8.75 | 84.69 | 95.51±1.79 | 29.51±8.75 | 38.85 | 49.67±1.79 |
| OPT-IML-Max-1.3B | 36.47 | 80.20±8.94 | 86.04 | 95.63±1.74 | 43.73±8.94 | 49.56 | 59.15±1.74 |
| OPT-6.7B | 6.33 | 85.14±4.57 | 61.79 | 97.42±2.32 | 78.81±4.57 | 55.46 | 91.09±2.32 |

Accuracy given 0.5% Ablation

| | Target Task | | Control Task | | Accuracy Differences | | |
|---|---|---|---|---|---|---|---|
| Model | Targeted Ablation (A) | Random Ablation (B) | Targeted Ablation (C) | Random Ablation (D) | (B) - (A) | (C) - (A) | (D) - (A) |
| GPT2-355M | 22.93 | 59.30±14.53 | 19.59 | 90.29±4.42 | 36.37±14.53 | -3.34 | 67.36±4.42 |
| OPT-1.3B | 31.31 | 62.74±15.78 | 85.03 | 91.69±6.99 | 31.43±15.78 | 53.72 | 60.38±6.99 |
| OPT-IML-1.3B | 35.57 | 65.01±9.55 | 79.09 | 91.02±4.48 | 29.44±9.55 | 43.51 | 55.45±4.48 |
| OPT-IML-Max-1.3B | 29.24 | 70.34±8.35 | 78.63 | 91.35±3.81 | 41.10±8.35 | 49.39 | 62.11±3.81 |
| OPT-6.7B | 0.66 | 74.93±8.49 | 47.38 | 94.85±4.79 | 74.27±8.49 | 46.72 | 94.19±4.79 |

Accuracy given 1.0% Ablation

| | Target Task | | Control Task | | Accuracy Differences | | |
|---|---|---|---|---|---|---|---|
| Model | Targeted Ablation (A) | Random Ablation (B) | Targeted Ablation (C) | Random Ablation (D) | (B) - (A) | (C) - (A) | (D) - (A) |
| GPT2-355M | 19.27 | 44.90±11.11 | 14.14 | 70.64±3.85 | 25.63±11.11 | -5.13 | 51.38±3.85 |
| OPT-1.3B | 16.25 | 44.03±18.32 | 70.22 | 81.04±19.49 | 27.79±18.32 | 53.98 | 64.79±19.49 |
| OPT-IML-1.3B | 15.47 | 44.68±14.04 | 68.29 | 81.93±10.04 | 29.21±14.04 | 52.83 | 66.47±10.04 |
| OPT-IML-Max-1.3B | 15.47 | 50.68±9.67 | 63.85 | 82.72±9.73 | 35.20±9.67 | 48.38 | 67.25±9.73 |
| OPT-6.7B | 1.81 | 62.89±7.96 | 37.77 | 93.94±2.48 | 61.08±7.96 | 35.96 | 92.13±2.48 |

Accuracy given 2.0% Ablation

Table 9: Results on the CEOs task with attention.

| Model | Target Task | | Control Task | | Accuracy Differences | | |
|---|---|---|---|---|---|---|---|
| | Targeted Ablation (A) | Random Ablation (B) | Targeted Ablation (C) | Random Ablation (D) | (B) - (A) | (C) - (A) | (D) - (A) |
| GPT2-355M | 79.49 | 98.30±1.02 | 93.22 | 98.33±1.78 | 18.82±1.02 | 13.73 | 18.84±1.78 |
| OPT-1.3B | 87.51 | 99.84±0.10 | 98.72 | 99.17±0.43 | 12.33±0.10 | 11.21 | 11.66±0.43 |
| OPT-IML-1.3B | 80.84 | 99.27±0.29 | 98.27 | 99.28±0.37 | 18.43±0.29 | 17.42 | 18.44±0.37 |
| OPT-IML-Max-1.3B | 85.23 | 99.37±0.27 | 98.16 | 99.33±0.28 | 14.14±0.27 | 12.93 | 14.10±0.28 |
| OPT-6.7B | 83.42 | 99.87±0.11 | 100.00 | 99.90±0.23 | 16.45±0.11 | 16.58 | 16.47±0.23 |
| GPT-Neox-20B | 50.60 | 92.49 | 69.03 | 97.18 | 41.89 | 18.44 | 46.59 |
| OPT-66B | 89.62 | 99.55 | 99.53 | 99.05 | 9.93 | 9.91 | 9.43 |
| Bloom-176B | 91.81 | 98.54 | 100.00 | 100.00 | 6.73 | 8.19 | 8.19 |

Accuracy given 0.1% Ablation

| Model | Target Task | | Control Task | | Accuracy Differences | | |
|---|---|---|---|---|---|---|---|
| | Targeted Ablation (A) | Random Ablation (B) | Targeted Ablation (C) | Random Ablation (D) | (B) - (A) | (C) - (A) | (D) - (A) |
| GPT2-355M | 75.30 | 95.61±1.09 | 92.39 | 95.32±1.62 | 20.31±1.09 | 17.09 | 20.01±1.62 |
| OPT-1.3B | 44.31 | 99.51±0.23 | 96.86 | 99.09±0.55 | 55.21±0.23 | 52.55 | 54.78±0.55 |
| OPT-IML-1.3B | 45.07 | 98.45±0.48 | 97.15 | 98.62±1.02 | 53.38±0.48 | 52.08 | 53.55±1.02 |
| OPT-IML-Max-1.3B | 44.39 | 98.48±0.25 | 95.79 | 98.18±0.60 | 54.09±0.25 | 51.39 | 53.79±0.60 |
| OPT-6.7B | 55.59 | 99.71±0.27 | 100.00 | 99.66±0.54 | 44.12±0.27 | 44.41 | 44.07±0.54 |
| GPT-Neox-20B | 52.57 | 89.33 | 60.57 | 96.24 | 36.76 | 8.00 | 43.68 |
| OPT-66B | 76.53 | 99.77 | 99.53 | 99.53 | 23.25 | 23.00 | 23.00 |
| Bloom-176B | 76.61 | 97.37 | 99.56 | 100.00 | 20.76 | 22.95 | 23.39 |

Accuracy given 0.5% Ablation

| Model | Target Task | | Control Task | | Accuracy Differences | | |
|---|---|---|---|---|---|---|---|
| | Targeted Ablation (A) | Random Ablation (B) | Targeted Ablation (C) | Random Ablation (D) | (B) - (A) | (C) - (A) | (D) - (A) |
| GPT2-355M | 68.61 | 91.59±3.75 | 89.10 | 94.93±3.21 | 22.98±3.75 | 20.49 | 26.32±3.21 |
| OPT-1.3B | 35.90 | 99.05±0.34 | 94.63 | 98.58±0.44 | 63.15±0.34 | 58.73 | 62.68±0.44 |
| OPT-IML-1.3B | 37.06 | 97.68±0.73 | 94.18 | 97.75±1.02 | 60.62±0.73 | 57.12 | 60.69±1.02 |
| OPT-IML-Max-1.3B | 36.94 | 97.87±0.47 | 92.74 | 97.60±0.78 | 60.93±0.47 | 55.80 | 60.65±0.78 |
| OPT-6.7B | 30.60 | 99.53±0.48 | 99.92 | 99.09±1.25 | 68.94±0.48 | 69.33 | 68.50±1.25 |
| GPT-Neox-20B | 49.41 | 87.35 | 56.34 | 96.24 | 37.94 | 6.93 | 46.84 |
| OPT-66B | 62.30 | 99.10 | 98.10 | 98.10 | 36.79 | 35.80 | 35.80 |
| Bloom-176B | 66.96 | 95.32 | 99.56 | 100.00 | 28.36 | 32.60 | 33.04 |

Accuracy given 1.0% Ablation

| Model | Target Task | | Control Task | | Accuracy Differences | | |
|---|---|---|---|---|---|---|---|
| | Targeted Ablation (A) | Random Ablation (B) | Targeted Ablation (C) | Random Ablation (D) | (B) - (A) | (C) - (A) | (D) - (A) |
| GPT2-355M | 62.60 | 87.32±3.49 | 87.14 | 93.40±3.09 | 24.73±3.49 | 24.54 | 30.80±3.09 |
| OPT-1.3B | 20.57 | 98.11±1.01 | 90.22 | 97.50±0.75 | 77.53±1.01 | 69.64 | 76.93±0.75 |
| OPT-IML-1.3B | 21.92 | 96.14±1.46 | 88.21 | 95.99±1.16 | 74.22±1.46 | 66.30 | 74.07±1.16 |
| OPT-IML-Max-1.3B | 23.83 | 96.67±1.53 | 86.71 | 96.45±1.17 | 72.84±1.53 | 62.88 | 72.62±1.17 |
| OPT-6.7B | 31.41 | 99.62±0.44 | 98.40 | 97.98±1.02 | 68.21±0.44 | 66.99 | 66.57±1.02 |
| GPT-Neox-20B | 45.65 | 87.65 | 50.12 | 96.01 | 42.00 | 4.47 | 50.36 |
| OPT-66B | 47.12 | 98.81 | 86.97 | 97.63 | 51.69 | 39.84 | 50.51 |
| Bloom-176B | 58.33 | 94.81 | 99.34 | 99.78 | 36.48 | 41.01 | 41.45 |

Accuracy given 2.0% Ablation

Table 10: Results on the Tickers task with MLP-Ks.

| Model | Target Task | | Control Task | | Accuracy Differences | | |
|---|---|---|---|---|---|---|---|
| | Targeted Ablation (A) | Random Ablation (B) | Targeted Ablation (C) | Random Ablation (D) | (B) - (A) | (C) - (A) | (D) - (A) |
| GPT2-355M | 91.42 | 98.47±0.92 | 98.33 | 98.32±1.56 | 7.04±0.92 | 6.91 | 6.90±1.56 |
| OPT-1.3B | 84.44 | 99.77±0.15 | 99.20 | 99.36±0.48 | 15.33±0.15 | 14.76 | 14.92±0.48 |
| OPT-IML-1.3B | 77.91 | 99.31±0.24 | 98.70 | 99.17±0.48 | 21.40±0.24 | 20.80 | 21.26±0.48 |
| OPT-IML-Max-1.3B | 81.31 | 99.42±0.09 | 99.09 | 99.30±0.38 | 18.12±0.09 | 17.79 | 17.99±0.38 |
| OPT-6.7B | 92.77 | 99.87±0.12 | 100.00 | 99.90±0.23 | 7.10±0.12 | 7.23 | 7.13±0.23 |

Accuracy given 0.1% Ablation

| Model | Target Task | | Control Task | | Accuracy Differences | | |
|---|---|---|---|---|---|---|---|
| | Targeted Ablation (A) | Random Ablation (B) | Targeted Ablation (C) | Random Ablation (D) | (B) - (A) | (C) - (A) | (D) - (A) |
| GPT2-355M | 90.62 | 95.12±2.68 | 94.99 | 96.15±1.38 | 4.50±2.68 | 4.37 | 5.53±1.38 |
| OPT-1.3B | 53.51 | 99.42±0.11 | 99.34 | 98.60±1.10 | 45.91±0.11 | 45.83 | 45.09±1.10 |
| OPT-IML-1.3B | 42.48 | 98.21±0.94 | 99.38 | 97.95±2.20 | 55.73±0.94 | 56.90 | 55.47±2.20 |
| OPT-IML-Max-1.3B | 43.00 | 98.44±0.24 | 98.16 | 98.07±0.77 | 55.45±0.24 | 55.16 | 55.07±0.77 |
| OPT-6.7B | 82.39 | 99.79±0.23 | 100.00 | 99.66±0.54 | 17.40±0.23 | 17.61 | 17.27±0.54 |

Accuracy given 0.5% Ablation

| Model | Target Task | | Control Task | | Accuracy Differences | | |
|---|---|---|---|---|---|---|---|
| | Targeted Ablation (A) | Random Ablation (B) | Targeted Ablation (C) | Random Ablation (D) | (B) - (A) | (C) - (A) | (D) - (A) |
| GPT2-355M | 84.82 | 92.24±4.17 | 95.73 | 93.81±2.47 | 7.42±4.17 | 10.92 | 8.99±2.47 |
| OPT-1.3B | 48.12 | 99.08±0.33 | 98.89 | 98.28±0.73 | 50.96±0.33 | 50.78 | 50.16±0.73 |
| OPT-IML-1.3B | 38.16 | 97.47±1.12 | 98.31 | 97.39±1.58 | 59.31±1.12 | 60.14 | 59.22±1.58 |
| OPT-IML-Max-1.3B | 39.44 | 97.94±0.30 | 97.42 | 97.80±0.56 | 58.50±0.30 | 57.98 | 58.36±0.56 |
| OPT-6.7B | 71.36 | 99.66±0.35 | 100.00 | 99.11±1.27 | 28.30±0.35 | 28.64 | 27.75±1.27 |

Accuracy given 1.0% Ablation

| Model | Target Task | | Control Task | | Accuracy Differences | | |
|---|---|---|---|---|---|---|---|
| | Targeted Ablation (A) | Random Ablation (B) | Targeted Ablation (C) | Random Ablation (D) | (B) - (A) | (C) - (A) | (D) - (A) |
| GPT2-355M | 65.42 | 87.10±4.61 | 79.18 | 92.34±2.63 | 21.68±4.61 | 13.77 | 26.92±2.63 |
| OPT-1.3B | 37.72 | 98.38±1.09 | 97.95 | 97.65±0.42 | 60.66±1.09 | 60.23 | 59.93±0.42 |
| OPT-IML-1.3B | 29.75 | 96.00±1.60 | 96.03 | 96.31±0.98 | 66.25±1.60 | 66.28 | 66.56±0.98 |
| OPT-IML-Max-1.3B | 32.83 | 96.96±0.79 | 95.91 | 97.32±0.81 | 64.13±0.79 | 63.08 | 64.49±0.81 |
| OPT-6.7B | 54.49 | 99.70±0.24 | 100.00 | 98.30±0.45 | 45.21±0.24 | 45.51 | 43.81±0.45 |

Accuracy given 2.0% Ablation

Table 11: Results on the Tickers task with MLP-Vs.

| | Target Task | | Control Task | | Accuracy Differences | | |
|---|---|---|---|---|---|---|---|
| Model | Targeted Ablation (A) | Random Ablation (B) | Targeted Ablation (C) | Random Ablation (D) | (B) - (A) | (C) - (A) | (D) - (A) |
| GPT2-355M | 81.34 | 92.91±1.40 | 79.83 | 96.84±1.23 | 11.57±1.40 | -1.51 | 15.50±1.23 |
| OPT-1.3B | 99.18 | 99.55±0.28 | 97.13 | 98.63±0.27 | 0.37±0.28 | -2.05 | -0.54±0.27 |
| OPT-IML-1.3B | 96.15 | 98.53±0.47 | 96.43 | 97.97±0.58 | 2.37±0.47 | 0.28 | 1.82±0.58 |
| OPT-IML-Max-1.3B | 97.73 | 98.36±0.53 | 96.83 | 98.24±0.41 | 0.63±0.53 | -0.90 | 0.51±0.41 |
| OPT-6.7B | 99.13 | 99.78±0.20 | 99.90 | 99.46±0.37 | 0.65±0.20 | 0.76 | 0.33±0.37 |

Accuracy given 0.1% Ablation

| | Target Task | | Control Task | | Accuracy Differences | | |
|---|---|---|---|---|---|---|---|
| Model | Targeted Ablation (A) | Random Ablation (B) | Targeted Ablation (C) | Random Ablation (D) | (B) - (A) | (C) - (A) | (D) - (A) |
| GPT2-355M | 56.10 | 82.95±4.98 | 16.62 | 94.13±2.44 | 26.85±4.98 | -39.48 | 38.03±2.44 |
| OPT-1.3B | 92.83 | 97.50±1.17 | 88.10 | 96.33±2.28 | 4.67±1.17 | -4.73 | 3.49±2.28 |
| OPT-IML-1.3B | 84.55 | 93.47±3.82 | 84.11 | 95.84±1.35 | 8.92±3.82 | -0.44 | 11.29±1.35 |
| OPT-IML-Max-1.3B | 86.11 | 96.38±1.84 | 86.12 | 94.79±3.36 | 10.27±1.84 | 0.01 | 8.68±3.36 |
| OPT-6.7B | 98.59 | 99.57±0.34 | 95.07 | 98.24±1.03 | 0.99±0.34 | -3.52 | -0.34±1.03 |

Accuracy given 0.5% Ablation

| | Target Task | | Control Task | | Accuracy Differences | | |
|---|---|---|---|---|---|---|---|
| Model | Targeted Ablation (A) | Random Ablation (B) | Targeted Ablation (C) | Random Ablation (D) | (B) - (A) | (C) - (A) | (D) - (A) |
| GPT2-355M | 47.30 | 69.85±8.32 | 12.80 | 88.99±6.71 | 22.56±8.32 | -34.50 | 41.69±6.71 |
| OPT-1.3B | 81.62 | 94.50±3.09 | 76.21 | 91.82±6.71 | 12.88±3.09 | -5.41 | 10.20±6.71 |
| OPT-IML-1.3B | 72.13 | 89.22±5.57 | 73.48 | 90.64±5.18 | 17.09±5.57 | 1.36 | 18.52±5.18 |
| OPT-IML-Max-1.3B | 73.74 | 91.72±4.23 | 77.00 | 90.43±5.48 | 17.97±4.23 | 3.26 | 16.68±5.48 |
| OPT-6.7B | 98.04 | 99.23±0.68 | 90.91 | 96.45±2.06 | 1.19±0.68 | -7.13 | -1.59±2.06 |

Accuracy given 1.0% Ablation

| | Target Task | | Control Task | | Accuracy Differences | | |
|---|---|---|---|---|---|---|---|
| Model | Targeted Ablation (A) | Random Ablation (B) | Targeted Ablation (C) | Random Ablation (D) | (B) - (A) | (C) - (A) | (D) - (A) |
| GPT2-355M | 35.86 | 53.42±7.23 | 9.64 | 69.53±6.12 | 17.56±7.23 | -26.22 | 33.67±6.12 |
| OPT-1.3B | 59.18 | 88.48±7.77 | 52.58 | 82.75±15.92 | 29.30±7.77 | -6.60 | 23.57±15.92 |
| OPT-IML-1.3B | 47.25 | 80.80±10.99 | 52.46 | 80.08±13.58 | 33.55±10.99 | 5.21 | 32.82±13.58 |
| OPT-IML-Max-1.3B | 49.11 | 82.32±10.88 | 58.91 | 81.68±11.25 | 33.21±10.88 | 9.80 | 32.57±11.25 |
| OPT-6.7B | 97.22 | 97.48±2.56 | 89.89 | 93.51±2.92 | 0.26±2.56 | -7.33 | -3.71±2.92 |

Accuracy given 2.0% Ablation

Table 12: Results on the Tickers task with attention.

| Model | Target Task | | Control Task | | Accuracy Differences | | |
|---|---|---|---|---|---|---|---|
| | Targeted Ablation (A) | Random Ablation (B) | Targeted Ablation (C) | Random Ablation (D) | (B) - (A) | (C) - (A) | (D) - (A) |
| GPT2-355M | 52.62 | 96.55±3.59 | 94.89 | 98.33±1.78 | 43.93±3.59 | 42.27 | 45.71±1.78 |
| OPT-1.3B | 58.29 | 96.43±1.73 | 99.05 | 99.39±0.52 | 38.14±1.73 | 40.76 | 41.11±0.52 |
| OPT-IML-1.3B | 60.66 | 98.48±1.38 | 98.43 | 99.40±0.28 | 37.82±1.38 | 37.77 | 38.74±0.28 |
| OPT-IML-Max-1.3B | 56.61 | 98.91±0.80 | 98.65 | 99.36±0.27 | 42.30±0.80 | 42.04 | 42.75±0.27 |
| OPT-6.7B | 68.34 | 98.75±1.24 | 100.00 | 99.80±0.28 | 30.41±1.24 | 31.66 | 31.46±0.28 |

Accuracy given 0.1% Ablation

| Model | Target Task | | Control Task | | Accuracy Differences | | |
|---|---|---|---|---|---|---|---|
| | Targeted Ablation (A) | Random Ablation (B) | Targeted Ablation (C) | Random Ablation (D) | (B) - (A) | (C) - (A) | (D) - (A) |
| GPT2-355M | 47.92 | 94.82±1.92 | 93.26 | 95.65±1.23 | 46.90±1.92 | 45.34 | 47.73±1.23 |
| OPT-1.3B | 13.47 | 95.14±2.55 | 96.22 | 99.10±0.56 | 81.68±2.55 | 82.75 | 85.63±0.56 |
| OPT-IML-1.3B | 14.94 | 95.36±1.98 | 97.11 | 98.74±0.92 | 80.42±1.98 | 82.17 | 83.80±0.92 |
| OPT-IML-Max-1.3B | 8.26 | 95.88±3.03 | 95.13 | 97.83±1.23 | 87.62±3.03 | 86.88 | 89.57±1.23 |
| OPT-6.7B | 30.27 | 97.44±0.91 | 100.00 | 99.57±0.51 | 67.17±0.91 | 69.73 | 69.29±0.51 |

Accuracy given 0.5% Ablation

| Model | Target Task | | Control Task | | Accuracy Differences | | |
|---|---|---|---|---|---|---|---|
| | Targeted Ablation (A) | Random Ablation (B) | Targeted Ablation (C) | Random Ablation (D) | (B) - (A) | (C) - (A) | (D) - (A) |
| GPT2-355M | 39.43 | 91.42±2.97 | 91.58 | 94.28±2.14 | 52.00±2.97 | 52.16 | 54.86±2.14 |
| OPT-1.3B | 13.19 | 93.96±1.49 | 96.93 | 98.64±0.44 | 80.77±1.49 | 83.74 | 85.45±0.44 |
| OPT-IML-1.3B | 12.13 | 94.24±2.35 | 96.55 | 97.95±0.89 | 82.10±2.35 | 84.42 | 85.82±0.89 |
| OPT-IML-Max-1.3B | 6.74 | 94.52±2.97 | 94.94 | 97.52±0.89 | 87.79±2.97 | 88.21 | 90.79±0.89 |
| OPT-6.7B | 16.99 | 95.86±1.37 | 99.97 | 98.90±1.17 | 78.87±1.37 | 82.98 | 81.91±1.17 |

Accuracy given 1.0% Ablation

| Model | Target Task | | Control Task | | Accuracy Differences | | |
|---|---|---|---|---|---|---|---|
| | Targeted Ablation (A) | Random Ablation (B) | Targeted Ablation (C) | Random Ablation (D) | (B) - (A) | (C) - (A) | (D) - (A) |
| GPT2-355M | 37.03 | 85.84±2.06 | 88.66 | 92.85±2.18 | 48.81±2.06 | 51.63 | 55.82±2.18 |
| OPT-1.3B | 12.52 | 91.57±1.64 | 98.50 | 97.69±0.53 | 79.05±1.64 | 85.98 | 85.17±0.53 |
| OPT-IML-1.3B | 6.67 | 92.05±3.60 | 95.41 | 96.36±1.12 | 85.38±3.60 | 88.75 | 89.69±1.12 |
| OPT-IML-Max-1.3B | 3.88 | 91.89±2.66 | 94.64 | 96.98±0.81 | 88.02±2.66 | 90.76 | 93.10±0.81 |
| OPT-6.7B | 10.61 | 93.52±1.22 | 99.47 | 98.19±0.97 | 82.91±1.22 | 88.85 | 87.58±0.97 |

Accuracy given 2.0% Ablation

Table 13: Results on the Directors task with MLP-Ks.

| Model | Target Task | | Control Task | | Accuracy Differences | | |
|---|---|---|---|---|---|---|---|
| | Targeted Ablation (A) | Random Ablation (B) | Targeted Ablation (C) | Random Ablation (D) | (B) - (A) | (C) - (A) | (D) - (A) |
| GPT2-355M | 56.82 | 96.55±3.50 | 100.00 | 98.66±1.52 | 39.73±3.50 | 43.18 | 41.84±1.52 |
| OPT-1.3B | 62.06 | 96.63±1.95 | 99.69 | 99.17±0.43 | 34.56±1.95 | 37.63 | 37.10±0.43 |
| OPT-IML-1.3B | 65.43 | 98.29±1.47 | 98.86 | 99.34±0.43 | 32.86±1.47 | 33.44 | 33.92±0.43 |
| OPT-IML-Max-1.3B | 70.57 | 99.13±0.61 | 98.79 | 99.27±0.42 | 28.57±0.61 | 28.23 | 28.70±0.42 |
| OPT-6.7B | 52.58 | 98.97±0.98 | 86.17 | 99.89±0.22 | 46.39±0.98 | 33.59 | 47.31±0.22 |

Accuracy given 0.1% Ablation

| Model | Target Task | | Control Task | | Accuracy Differences | | |
|---|---|---|---|---|---|---|---|
| | Targeted Ablation (A) | Random Ablation (B) | Targeted Ablation (C) | Random Ablation (D) | (B) - (A) | (C) - (A) | (D) - (A) |
| GPT2-355M | 48.01 | 93.97±2.31 | 99.17 | 95.49±1.39 | 45.96±2.31 | 51.16 | 47.48±1.39 |
| OPT-1.3B | 19.65 | 94.39±1.73 | 98.10 | 98.85±0.68 | 74.74±1.73 | 78.45 | 79.20±0.68 |
| OPT-IML-1.3B | 27.54 | 95.60±2.20 | 95.99 | 98.73±0.93 | 68.06±2.20 | 68.46 | 71.19±0.93 |
| OPT-IML-Max-1.3B | 28.70 | 95.91±3.04 | 95.15 | 98.18±0.60 | 67.21±3.04 | 66.46 | 69.49±0.60 |
| OPT-6.7B | 28.52 | 97.55±0.84 | 78.19 | 99.55±0.51 | 69.03±0.84 | 49.67 | 71.03±0.51 |

Accuracy given 0.5% Ablation

| Model | Target Task | | Control Task | | Accuracy Differences | | |
|---|---|---|---|---|---|---|---|
| | Targeted Ablation (A) | Random Ablation (B) | Targeted Ablation (C) | Random Ablation (D) | (B) - (A) | (C) - (A) | (D) - (A) |
| GPT2-355M | 35.23 | 89.71±4.80 | 95.06 | 94.28±2.93 | 54.49±4.80 | 59.83 | 59.05±2.93 |
| OPT-1.3B | 15.33 | 93.54±0.76 | 96.99 | 98.44±0.42 | 78.21±0.76 | 81.66 | 83.11±0.42 |
| OPT-IML-1.3B | 22.09 | 93.65±2.15 | 95.53 | 97.81±0.96 | 71.56±2.15 | 73.44 | 75.72±0.96 |
| OPT-IML-Max-1.3B | 22.85 | 94.47±2.92 | 94.20 | 97.57±0.91 | 71.61±2.92 | 71.35 | 74.71±0.91 |
| OPT-6.7B | 7.07 | 95.91±0.90 | 66.15 | 99.00±1.20 | 88.84±0.90 | 59.08 | 91.93±1.20 |

Accuracy given 1.0% Ablation

| Model | Target Task | | Control Task | | Accuracy Differences | | |
|---|---|---|---|---|---|---|---|
| | Targeted Ablation (A) | Random Ablation (B) | Targeted Ablation (C) | Random Ablation (D) | (B) - (A) | (C) - (A) | (D) - (A) |
| GPT2-355M | 29.53 | 84.73±3.06 | 80.79 | 92.52±2.73 | 55.20±3.06 | 51.26 | 62.99±2.73 |
| OPT-1.3B | 8.02 | 91.95±1.88 | 94.79 | 97.62±0.42 | 83.92±1.88 | 86.77 | 89.60±0.42 |
| OPT-IML-1.3B | 11.78 | 89.76±3.56 | 94.68 | 95.93±1.11 | 77.99±3.56 | 82.91 | 84.16±1.11 |
| OPT-IML-Max-1.3B | 12.56 | 91.72±2.68 | 92.39 | 96.36±1.71 | 79.16±2.68 | 79.83 | 83.80±1.71 |
| OPT-6.7B | 3.35 | 92.40±3.02 | 59.57 | 98.19±0.48 | 89.05±3.02 | 56.22 | 94.84±0.48 |

Accuracy given 2.0% Ablation

Table 14: Results on the Directors task with MLP-Vs.

| Model | Target Task | | Control Task | | Accuracy Differences | | |
|---|---|---|---|---|---|---|---|
| | Targeted Ablation (A) | Random Ablation (B) | Targeted Ablation (C) | Random Ablation (D) | (B) - (A) | (C) - (A) | (D) - (A) |
| GPT2-355M | 52.47 | 93.13±4.85 | 81.53 | 96.33±1.39 | 40.66±4.85 | 29.06 | 43.86±1.39 |
| OPT-1.3B | 93.01 | 95.87±1.29 | 98.41 | 98.32±0.80 | 2.86±1.29 | 5.40 | 5.31±0.80 |
| OPT-IML-1.3B | 91.06 | 93.70±2.98 | 98.70 | 97.56±1.31 | 2.64±2.98 | 7.64 | 6.50±1.31 |
| OPT-IML-Max-1.3B | 95.43 | 96.22±1.81 | 98.65 | 98.48±0.24 | 0.79±1.81 | 3.22 | 3.06±0.24 |
| OPT-6.7B | 87.33 | 97.47±1.07 | 96.95 | 99.42±0.38 | 10.14±1.07 | 9.62 | 12.09±0.38 |

Accuracy given 0.1% Ablation

| Model | Target Task | | Control Task | | Accuracy Differences | | |
|---|---|---|---|---|---|---|---|
| | Targeted Ablation (A) | Random Ablation (B) | Targeted Ablation (C) | Random Ablation (D) | (B) - (A) | (C) - (A) | (D) - (A) |
| GPT2-355M | 9.62 | 87.94±3.63 | 44.03 | 92.62±3.75 | 78.32±3.63 | 34.41 | 83.00±3.75 |
| OPT-1.3B | 78.56 | 86.89±2.15 | 94.34 | 96.56±1.80 | 8.32±2.15 | 15.78 | 17.99±1.80 |
| OPT-IML-1.3B | 82.97 | 86.26±4.95 | 91.01 | 96.28±1.28 | 3.29±4.95 | 8.04 | 13.31±1.28 |
| OPT-IML-Max-1.3B | 80.51 | 85.31±5.46 | 88.61 | 94.79±3.34 | 4.80±5.46 | 8.11 | 14.29±3.34 |
| OPT-6.7B | 69.24 | 93.31±2.56 | 83.11 | 97.52±2.13 | 24.06±2.56 | 13.86 | 28.27±2.13 |

Accuracy given 0.5% Ablation

| Model | Target Task | | Control Task | | Accuracy Differences | | |
|---|---|---|---|---|---|---|---|
| | Targeted Ablation (A) | Random Ablation (B) | Targeted Ablation (C) | Random Ablation (D) | (B) - (A) | (C) - (A) | (D) - (A) |
| GPT2-355M | 0.29 | 76.07±10.65 | 23.57 | 91.07±3.51 | 75.77±10.65 | 23.28 | 90.77±3.51 |
| OPT-1.3B | 64.34 | 75.59±6.11 | 88.81 | 93.78±2.98 | 11.25±6.11 | 24.47 | 29.44±2.98 |
| OPT-IML-1.3B | 68.40 | 75.44±8.55 | 86.70 | 89.93±6.58 | 7.03±8.55 | 18.30 | 21.53±6.58 |
| OPT-IML-Max-1.3B | 72.09 | 75.90±8.55 | 84.94 | 91.26±3.95 | 3.82±8.55 | 12.85 | 19.17±3.95 |
| OPT-6.7B | 52.55 | 88.98±3.83 | 73.65 | 95.92±2.74 | 36.44±3.83 | 21.11 | 43.38±2.74 |

Accuracy given 1.0% Ablation

| Model | Target Task | | Control Task | | Accuracy Differences | | |
|---|---|---|---|---|---|---|---|
| | Targeted Ablation (A) | Random Ablation (B) | Targeted Ablation (C) | Random Ablation (D) | (B) - (A) | (C) - (A) | (D) - (A) |
| GPT2-355M | 0.00 | 58.22±8.37 | 17.36 | 71.82±2.05 | 58.22±8.37 | 17.36 | 71.82±2.05 |
| OPT-1.3B | 35.78 | 52.88±14.50 | 77.78 | 88.17±6.78 | 17.10±14.50 | 42.00 | 52.39±6.78 |
| OPT-IML-1.3B | 39.28 | 53.59±17.81 | 78.46 | 76.98±20.01 | 14.32±17.81 | 39.19 | 37.70±20.01 |
| OPT-IML-Max-1.3B | 55.51 | 57.16±14.49 | 78.00 | 84.20±8.25 | 1.64±14.49 | 22.49 | 28.69±8.25 |
| OPT-6.7B | 31.28 | 81.12±6.64 | 68.09 | 93.19±3.39 | 49.83±6.64 | 36.80 | 61.91±3.39 |

Accuracy given 2.0% Ablation

Table 15: Results on the Directors task with attention.

| Model | Target Task | | Control Task | | Accuracy Differences | | |
|---|---|---|---|---|---|---|---|
| | Targeted Ablation (A) | Random Ablation (B) | Targeted Ablation (C) | Random Ablation (D) | (B) - (A) | (C) - (A) | (D) - (A) |
| GPT2-355M | 100.00 | 100.00±0.00 | 94.10 | 98.49±1.82 | 0.00±0.00 | -5.90 | -1.51±1.82 |
| OPT-1.3B | 84.62 | 99.26±1.02 | 99.85 | 99.30±0.42 | 14.64±1.02 | 15.23 | 14.68±0.42 |
| OPT-IML-1.3B | 84.23 | 99.31±0.94 | 99.56 | 99.19±0.52 | 15.08±0.94 | 15.33 | 14.96±0.52 |
| OPT-IML-Max-1.3B | 70.34 | 94.02±5.35 | 98.62 | 99.12±0.60 | 23.68±5.35 | 28.29 | 28.78±0.60 |
| OPT-6.7B | 71.68 | 99.44±1.17 | 100.00 | 99.89±0.22 | 27.76±1.17 | 28.32 | 28.21±0.22 |

Accuracy given 0.1% Ablation

| Model | Target Task | | Control Task | | Accuracy Differences | | |
|---|---|---|---|---|---|---|---|
| | Targeted Ablation (A) | Random Ablation (B) | Targeted Ablation (C) | Random Ablation (D) | (B) - (A) | (C) - (A) | (D) - (A) |
| GPT2-355M | 86.02 | 100.00±0.00 | 94.07 | 96.15±1.39 | 13.98±0.00 | 8.05 | 10.13±1.39 |
| OPT-1.3B | 47.46 | 95.44±4.17 | 98.74 | 98.73±0.86 | 47.98±4.17 | 51.28 | 51.27±0.86 |
| OPT-IML-1.3B | 70.94 | 90.24±3.80 | 99.44 | 98.96±0.93 | 19.30±3.80 | 28.50 | 28.01±0.93 |
| OPT-IML-Max-1.3B | 40.00 | 89.42±5.79 | 97.57 | 98.41±0.54 | 49.42±5.79 | 57.57 | 58.41±0.54 |
| OPT-6.7B | 54.36 | 97.87±1.00 | 99.78 | 99.44±0.58 | 43.51±1.00 | 45.43 | 45.09±0.58 |

Accuracy given 0.5% Ablation

| Model | Target Task | | Control Task | | Accuracy Differences | | |
|---|---|---|---|---|---|---|---|
| | Targeted Ablation (A) | Random Ablation (B) | Targeted Ablation (C) | Random Ablation (D) | (B) - (A) | (C) - (A) | (D) - (A) |
| GPT2-355M | 71.91 | 91.72±7.56 | 93.25 | 94.46±2.40 | 19.81±7.56 | 21.34 | 22.55±2.40 |
| OPT-1.3B | 38.72 | 93.90±4.95 | 98.56 | 98.25±0.91 | 55.18±4.95 | 59.84 | 59.53±0.91 |
| OPT-IML-1.3B | 59.35 | 87.24±3.43 | 98.31 | 97.98±0.89 | 27.89±3.43 | 38.95 | 38.63±0.89 |
| OPT-IML-Max-1.3B | 36.78 | 89.33±6.07 | 96.81 | 97.66±0.69 | 52.55±6.07 | 60.03 | 60.88±0.69 |
| OPT-6.7B | 44.44 | 95.64±2.36 | 99.47 | 98.91±1.17 | 51.19±2.36 | 55.02 | 54.46±1.17 |

Accuracy given 1.0% Ablation

| Model | Target Task | | Control Task | | Accuracy Differences | | |
|---|---|---|---|---|---|---|---|
| | Targeted Ablation (A) | Random Ablation (B) | Targeted Ablation (C) | Random Ablation (D) | (B) - (A) | (C) - (A) | (D) - (A) |
| GPT2-355M | 61.08 | 86.60±4.74 | 91.38 | 93.01±2.43 | 25.52±4.74 | 30.30 | 31.94±2.43 |
| OPT-1.3B | 23.60 | 90.87±7.28 | 98.25 | 97.31±1.11 | 67.27±7.28 | 74.65 | 73.71±1.11 |
| OPT-IML-1.3B | 34.78 | 81.62±5.34 | 96.03 | 95.99±1.16 | 46.85±5.34 | 61.26 | 61.22±1.16 |
| OPT-IML-Max-1.3B | 30.23 | 89.33±6.89 | 95.30 | 96.16±1.73 | 59.11±6.89 | 65.07 | 65.93±1.73 |
| OPT-6.7B | 44.44 | 97.22±0.00 | 99.47 | 98.30±1.02 | 52.78±0.00 | 55.02 | 53.85±1.02 |

Accuracy given 2.0% Ablation

Table 16: Results on the Authors task with MLP-Ks.

| Model | Target Task | | Control Task | | Accuracy Differences | | |
|---|---|---|---|---|---|---|---|
| | Targeted Ablation (A) | Random Ablation (B) | Targeted Ablation (C) | Random Ablation (D) | (B) - (A) | (C) - (A) | (D) - (A) |
| GPT2-355M | 100.00 | 100.00±0.00 | 99.18 | 98.66±1.52 | 0.00±0.00 | -0.82 | -1.34±1.52 |
| OPT-1.3B | 57.82 | 99.26±1.02 | 99.03 | 99.04±0.60 | 41.43±1.02 | 41.20 | 41.22±0.60 |
| OPT-IML-1.3B | 60.84 | 98.27±2.11 | 99.00 | 99.22±0.43 | 37.44±2.11 | 38.17 | 38.39±0.43 |
| OPT-IML-Max-1.3B | 71.72 | 94.34±5.72 | 99.09 | 99.30±0.38 | 22.62±5.72 | 27.37 | 27.58±0.38 |
| OPT-6.7B | 57.93 | 99.47±1.18 | 87.26 | 99.90±0.23 | 41.54±1.18 | 29.33 | 41.97±0.23 |

Accuracy given 0.1% Ablation

| Model | Target Task | | Control Task | | Accuracy Differences | | |
|---|---|---|---|---|---|---|---|
| | Targeted Ablation (A) | Random Ablation (B) | Targeted Ablation (C) | Random Ablation (D) | (B) - (A) | (C) - (A) | (D) - (A) |
| GPT2-355M | 86.02 | 100.00±0.00 | 96.66 | 96.32±1.62 | 13.98±0.00 | 10.65 | 10.30±1.62 |
| OPT-1.3B | 46.15 | 95.44±4.17 | 99.38 | 98.85±0.68 | 49.28±4.17 | 53.23 | 52.69±0.68 |
| OPT-IML-1.3B | 50.00 | 90.10±3.68 | 97.13 | 98.62±1.02 | 40.10±3.68 | 47.13 | 48.62±1.02 |
| OPT-IML-Max-1.3B | 60.23 | 92.05±7.16 | 96.38 | 98.18±0.61 | 31.82±7.16 | 36.16 | 37.95±0.61 |
| OPT-6.7B | 43.24 | 98.65±0.94 | 81.30 | 99.66±0.54 | 55.40±0.94 | 38.05 | 56.41±0.54 |

Accuracy given 0.5% Ablation

| Model | Target Task | | Control Task | | Accuracy Differences | | |
|---|---|---|---|---|---|---|---|
| | Targeted Ablation (A) | Random Ablation (B) | Targeted Ablation (C) | Random Ablation (D) | (B) - (A) | (C) - (A) | (D) - (A) |
| GPT2-355M | 71.91 | 94.48±7.56 | 97.43 | 93.98±2.60 | 22.57±7.56 | 25.52 | 22.07±2.60 |
| OPT-1.3B | 42.44 | 93.16±4.62 | 98.43 | 98.41±0.47 | 50.72±4.62 | 55.99 | 55.97±0.47 |
| OPT-IML-1.3B | 43.10 | 86.20±3.43 | 95.71 | 97.75±1.02 | 43.11±3.43 | 52.61 | 54.65±1.02 |
| OPT-IML-Max-1.3B | 51.95 | 90.39±6.63 | 95.13 | 97.66±0.76 | 38.44±6.63 | 43.18 | 45.71±0.76 |
| OPT-6.7B | 32.39 | 96.59±2.02 | 71.58 | 99.10±1.26 | 64.20±2.02 | 39.19 | 66.71±1.26 |

Accuracy given 1.0% Ablation

| Model | Target Task | | Control Task | | Accuracy Differences | | |
|---|---|---|---|---|---|---|---|
| | Targeted Ablation (A) | Random Ablation (B) | Targeted Ablation (C) | Random Ablation (D) | (B) - (A) | (C) - (A) | (D) - (A) |
| GPT2-355M | 57.63 | 91.53±8.57 | 88.45 | 92.79±2.97 | 33.90±8.57 | 30.82 | 35.17±2.97 |
| OPT-1.3B | 34.88 | 88.62±5.73 | 96.54 | 97.53±0.48 | 53.74±5.73 | 61.66 | 62.65±0.48 |
| OPT-IML-1.3B | 29.06 | 78.48±5.98 | 92.87 | 95.99±1.06 | 49.42±5.98 | 63.82 | 66.93±1.06 |
| OPT-IML-Max-1.3B | 35.57 | 87.11±5.72 | 92.72 | 96.63±1.16 | 51.55±5.72 | 57.16 | 61.07±1.16 |
| OPT-6.7B | 13.89 | 95.00±4.97 | 67.02 | 98.09±0.61 | 81.11±4.97 | 53.13 | 84.20±0.61 |

Accuracy given 2.0% Ablation

Table 17: Results on the Authors task with MLP-Vs.

| Model | Target Task | | Control Task | | Accuracy Differences | | |
|---|---|---|---|---|---|---|---|
| | Targeted Ablation (A) | Random Ablation (B) | Targeted Ablation (C) | Random Ablation (D) | (B) - (A) | (C) - (A) | (D) - (A) |
| GPT2-355M | 85.71 | 100.00±0.00 | 73.26 | 96.83±1.22 | 14.29±0.00 | -12.46 | 11.12±1.22 |
| OPT-1.3B | 90.45 | 95.81±4.95 | 99.20 | 98.57±0.33 | 5.36±4.95 | 8.75 | 8.12±0.33 |
| OPT-IML-1.3B | 89.41 | 91.48±7.60 | 98.15 | 97.59±1.25 | 2.07±7.60 | 8.74 | 8.19±1.25 |
| OPT-IML-Max-1.3B | 93.33 | 92.05±4.32 | 96.95 | 98.61±0.48 | -1.29±4.32 | 3.62 | 5.27±0.48 |
| OPT-6.7B | 83.06 | 97.86±1.20 | 90.88 | 99.36±0.44 | 14.80±1.20 | 7.82 | 16.29±0.44 |

Accuracy given 0.1% Ablation

| Model | Target Task | | Control Task | | Accuracy Differences | | |
|---|---|---|---|---|---|---|---|
| | Targeted Ablation (A) | Random Ablation (B) | Targeted Ablation (C) | Random Ablation (D) | (B) - (A) | (C) - (A) | (D) - (A) |
| GPT2-355M | 1.82 | 97.20±6.25 | 42.19 | 92.96±3.20 | 95.39±6.25 | 40.37 | 91.15±3.20 |
| OPT-1.3B | 62.32 | 84.88±5.44 | 98.10 | 96.70±1.52 | 22.56±5.44 | 35.78 | 34.38±1.52 |
| OPT-IML-1.3B | 50.97 | 84.38±9.24 | 91.49 | 95.28±2.21 | 33.41±9.24 | 40.52 | 44.31±2.21 |
| OPT-IML-Max-1.3B | 54.69 | 73.83±17.74 | 93.31 | 94.80±3.33 | 19.14±17.74 | 38.62 | 40.11±3.33 |
| OPT-6.7B | 62.01 | 96.29±1.09 | 84.19 | 98.05±1.22 | 34.28±1.09 | 22.17 | 36.03±1.22 |

Accuracy given 0.5% Ablation

| Model | Target Task | | Control Task | | Accuracy Differences | | |
|---|---|---|---|---|---|---|---|
| | Targeted Ablation (A) | Random Ablation (B) | Targeted Ablation (C) | Random Ablation (D) | (B) - (A) | (C) - (A) | (D) - (A) |
| GPT2-355M | 0.00 | 86.10±24.11 | 24.33 | 90.26±4.46 | 86.10±24.11 | 24.33 | 90.26±4.46 |
| OPT-1.3B | 50.39 | 72.35±7.28 | 93.43 | 93.51±3.41 | 21.96±7.28 | 43.04 | 43.12±3.41 |
| OPT-IML-1.3B | 43.10 | 73.24±11.51 | 89.37 | 90.88±4.75 | 30.15±11.51 | 46.28 | 47.78±4.75 |
| OPT-IML-Max-1.3B | 40.45 | 61.74±18.00 | 88.81 | 90.01±6.30 | 21.29±18.00 | 48.35 | 49.56±6.30 |
| OPT-6.7B | 38.21 | 91.34±1.93 | 76.13 | 96.14±2.42 | 53.13±1.93 | 37.92 | 57.92±2.42 |

Accuracy given 1.0% Ablation

| Model | Target Task | | Control Task | | Accuracy Differences | | |
|---|---|---|---|---|---|---|---|
| | Targeted Ablation (A) | Random Ablation (B) | Targeted Ablation (C) | Random Ablation (D) | (B) - (A) | (C) - (A) | (D) - (A) |
| GPT2-355M | 0.00 | 65.01±18.77 | 18.00 | 70.56±4.01 | 65.01±18.77 | 18.00 | 70.56±4.01 |
| OPT-1.3B | 27.71 | 47.40±15.68 | 83.98 | 87.08±8.01 | 19.69±15.68 | 56.27 | 59.37±8.01 |
| OPT-IML-1.3B | 29.06 | 50.78±17.19 | 85.40 | 81.96±10.00 | 21.72±17.19 | 56.34 | 52.90±10.00 |
| OPT-IML-Max-1.3B | 14.24 | 38.15±20.30 | 79.76 | 80.42±13.40 | 23.91±20.30 | 65.52 | 66.19±13.40 |
| OPT-6.7B | 25.00 | 85.00±5.05 | 67.02 | 93.40±3.07 | 60.00±5.05 | 42.02 | 68.40±3.07 |

Accuracy given 2.0% Ablation

Table 18: Results on the Authors task with attention.