# OpenReview forum: "Can We Statically Locate Knowledge in Large Language Models? Financial Domain and Toxicity Reduction Case Studies"
_EMNLP/2024/Workshop/BlackBoxNLP — BlackboxNLP 2024_

### Official Review · Reviewer_Ffcq · 2024-08-28

**Overall Assessment:** 4
**Confidence:** 4

**Best Paper:**

1

**Best Paper Justification:**

N/A

**Comments Questions Suggestions And Typos:**

Overall, this research offers a promising approach to statically locating knowledge in LLMs, paving the way for further exploration of how these models encode and store information. Additionally, I appreciate the authors' clear presentation and the easy-to-follow explanations of their experimental setup and methodology.

**Paper Summary:**

This paper introduces a novel method for statically identifying where specific knowledge is stored within the parameters of a large language model (LLM). The method leverages the idea that model parameters can be interpreted as embeddings in the same embedding space. By searching for parameters most similar (in embedding space) to a query representing the desired knowledge, the authors are able to locate relevant parameters without requiring forward or backward passes.

**Summary Of Strengths:**

- The static knowledge localization method represents a significant shift from traditional dynamic analysis techniques, which typically depend on input-based evaluations.
- The ability to locate and potentially modify specific knowledge within LLMs holds considerable practical value for areas like content creation and reducing bias.
- The authors carried out an extensive evaluation across various tasks, model architectures, and ablation strategies.
- The experiments clearly showed the effectiveness of the method. It was tested on financial information tasks (e.g., CEOs, tickers) to assess its ability to identify parameters that store relevant knowledge. Models performed considerably worse when the parameters identified were zeroed out.
- The method was also used to identify and remove model parameters linked to generating toxic language. "the toxicity of the models, with a drop of more than 35% when only1% of the units are ablated."

**Summary Of Weaknesses:**

Given the promising results in the financial domain and toxicity cases, it would be exciting to see how well the method performs in other domains and tasks to further explore its generalizability.

---

### Official Review · Reviewer_6p8u · 2024-09-08

**Overall Assessment:** 4
**Confidence:** 4

**Best Paper:**

1

**Best Paper Justification:**

N/A

**Comments Questions Suggestions And Typos:**

Any reasoning behind picking financial domain and toxicity specifically for experimentation?

**Paper Summary:**

The paper focuses on identifying which parameters contribute to specific knowledge/ behaviors in the LLM. They use static analysis to project weights into token space and computing embedding similarity between tokens and weights to isolate weight specific behaviors.

**Summary Of Strengths:**

The paper is well written and has depth of experimentation. Gives unique insights into interpreting neural networks.

**Summary Of Weaknesses:**

Experimentation could be done across domains/tasks to analyze the generalizability of the proposed approach,

---

### Official Review · Reviewer_y6XL · 2024-09-10

**Overall Assessment:** 4
**Confidence:** 3

**Best Paper:**

1

**Best Paper Justification:**

NA

**Comments Questions Suggestions And Typos:**

- I think the argument in lines 36-39 is problematic. Authors cite "we don't know what to ask" as an issue with existing probing methods, but this paper also uses a 'query' to look for knowledge in the parameters. Am I missing something here?
- Is there any difference with the approach of Dar et al., 2022? Section 2.1 seem to indicate its the same.
- The legend for control task in Figure 1 is unclear.
- What are $n$ and $N$ in Figure 2 caption? Also, are $k$ and $K$ different in lines 271-275? $k$ is not used anywhere else in the text, so you could skip defining a variable for this.
- OPT-66B presents interesting behavior on the control task in Table 1. Why is there a significant drop in performance with 0.1% ablation on the control task? Did you happen to investigate this?
- From Table 2, I couldn’t find a consistent pattern amongst the layers. Does this imply different models store ‘CEO’ knowledge in different layers? Some discussion on this can be helpful.
- To be consistent with Figure 3, I would recommend using % ablated units as x-axis in Figure 1.
- I couldn’t follow the reasoning in first paragraph of section 4.2, specifically about Bloom-176B and GPT2-355M. The explanation about model size extremities only holds for ‘CEOs’ category. From Figure 3 (’Tickers’), all OPT models seem to show similar drops as Bloom-176B. So, the argument about model size extremities doesn’t seem to hold. Also, any reason for not report Bloom-176B for ‘Directors’?

**Paper Summary:**

For a given query (/question), this paper studies techniques to localize the knowledge (/answer) to specific parameters in a transformer. Using techniques from prior work, they project the parameters (mostly MLP) into the embedding space and then perform kNN search between the query embedding and parameter embeddings. They evaluate the impact of these identified parameters by zeroing them out and comparing it against a control setting. On multiple zero-shot question-answering tasks, author(s) show the effectiveness of knowledge localization on variants of GPT-2, GPT-Neo, OPT and Bloom models.

Results show interesting trends for tasks such as company name --> CEO name prediction, but results are mixed for other tasks such as tickers, authors, directors etc.,

**Summary Of Strengths:**

- The experimental setup is quite exhaustive and addresses a variety of research questions about knowledge localization. They also show a downstream application in toxicity reduction by zeroing out parameters that are most similar to the toxic queries.
- The paper covers multiple model families (GPT-2, OPT and Bloom), and models at different scales. The paper studies the effects across model families and sizes.
- The simplicity of the methodology can potentially allow extension to other tasks and domains at the same model scale.

**Summary Of Weaknesses:**

- The current paper overtly focuses on the CEO task in the main paper text. As the author(s) hint in section 4.3, the generalization to other tasks is not guaranteed. On tasks such tickers, and directors the differences between targeted and controlled ablations are less pronounced (Table 3, Figure 3).
- Some parts of the analysis are quite hard to follow, specifically around layer-wise distributions, ablations on tickers and directors tasks. The paper can benefit from editing parts of section 4 to make it easier to understand the contributions. In the current form, author(s) refer back-and-forth between many figures and tables. See my questions below.
- As author(s) acknowledge, MLP seems more useful than attention layers (lines 437-450). I think this should be mentioned at the start of section 4 to set expectations for the readers. This is important because sections 2 and 3 do not make this distinction.

---

### Decision · Program_Chairs · 2024-09-19

**Decision:**

Accept

**Comment:**

Interesting static analysis of models on multiple tasks